# A cross-sectional study of functional and metabolic changes during aging through the lifespan in male mice

Michael A Petr[1,2], Irene Alfaras[2†], Melissa Krawcyzk[3], Woei-Nan Bair[2,4], Sarah J Mitchell[2‡], Christopher H Morrell[3], Stephanie A Studenski[2,5], Nathan L Price[2,6,7,8,9], Kenneth W Fishbein[10], Richard G Spencer[10], Morten Scheibye-Knudsen[1], Edward G Lakatta[3], Luigi Ferrucci[2], Miguel A Aon[2,3], Michel Bernier[2], Rafael de Cabo[2*]

[1]Center for Healthy Aging, ICMM, University of Copenhagen, Copenhagen, Denmark; [2]Translational Gerontology Branch, National Institute on Aging, NIH, Baltimore, United States; [3]Laboratory of Cardiovascular Science, National Institute on Aging, NIH, Baltimore, United States; [4]Department of Physical Therapy, University of Sciences, Philadelphia, United States; [5]Division of Geriatric Medicine, Department of Medicine, University of Pittsburgh, Pittsburgh, United States; [6]Integrative Cell Signaling and Neurobiology of Metabolism Program, Yale University School of Medicine, New Haven, United States; [7]Vascular Biology and Therapeutics Program, Yale University School of Medicine, New Haven , United States ; [8]Department of Comparative Medicine, Yale University School of Medicine, New Haven, United States ; [9]Department of Pathology, Yale University School of Medicine, New Haven,  United States ; [10]Laboratory of Clinical Investigation, National Institute on Aging, NIH, Baltimore, United States

*For correspondence:
decabora@grc.nia.nih.gov

Present address: †Aging Institute, University of Pittsburgh School of Medicine, Pittsburgh, United States; ‡Department of Health Sciences and Technology, ETH Zürich, Zürich, Switzerland

Competing interests: The authors declare that no competing interests exist.

**Abstract** Aging is associated with distinct phenotypical, physiological, and functional changes, leading to disease and death. The progression of aging-related traits varies widely among individuals, influenced by their environment, lifestyle, and genetics. In this study, we conducted physiologic and functional tests cross-sectionally throughout the entire lifespan of male C57BL/6N mice. In parallel, metabolomics analyses in serum, brain, liver, heart, and skeletal muscle were also performed to identify signatures associated with frailty and age-dependent functional decline. Our findings indicate that declines in gait speed as a function of age and frailty are associated with a dramatic increase in the energetic cost of physical activity and decreases in working capacity. Aging and functional decline prompt organs to rewire their metabolism and substrate selection and toward redox-related pathways, mainly in liver and heart. Collectively, the data provide a framework to further understand and characterize processes of aging at the individual organism and organ levels.

## Introduction

Genetics, lifestyle, and environmental factors contribute to different rates of aging. Across the lifespan, most species undergo complex phenotypical and functional changes that affect all organs of the body and the systems that allow communication and coordination between tissues. While this concept of aging has existed for a very long time, several milestones have recently been achieved in aging research, both in terms of the identification and modulation of pathways critical to aging in model organisms, and the characterization of factors relevant to human aging. In particular,

important progress has been made in comparing the phenotypes and rate of decline due to chronological aging *versus* disease-related factors, the relationship between health risk factors and aging, and the impact of behavioral trends on health and risk of disease promotion in humans (Healthy Aging: Lessons from the Baltimore Longitudinal Study of Aging, NIA, NIH, US DHHS, July 2010). There is significant variability in survival in the older population that cannot be explained solely by age and sex alone.

Under well-controlled environmental conditions, laboratory mice can live up to 4 years (*Miller et al., 2002*). A wide range of physiological, functional, behavioral, and pathological changes occur with age that determine how long mice live. Mice develop many of the histopathological characteristics of human aging during their lifespan, such as lymphocyte infiltration, tissue inflammation and necrosis, cancer, and amyloidosis (*Mitchell et al., 2016*; *Mitchell et al., 2019*). Understanding the process of normal aging in mice and identifying murine aging predictors and biomarkers will aid in the development of interventions capable of improving health and quality of life that are translatable to humans.

Research on aging is shifting its focus toward understanding the underlying differences in the aging phenotype of *chronological* and *biological* aging. Resilience and associated compensatory homeostatic mechanisms slow functional decline, resulting in great variability in survival (*Ferrucci et al., 2020*). For instance, physical performance metrics have provided important information about individualized estimates of survival as evidenced by the association of age-related decline in gait speed with reduced overall survival in humans (*Studenski et al., 2011*; *White et al., 2013*). However, clinically detectable changes currently utilized to characterize the aging phenotype are only apparent when compensatory mechanisms begin to fail (*Ferrucci et al., 2020*). In this context, the nature of the processes (e.g. metabolic, immunologic) and underlying mechanisms causing these compensatory maladaptations are not well understood (*Ferrucci et al., 2020*; *Adelnia et al., 2020*). All hallmarks of aging exhibit prominent anabolic and catabolic effects (*López-Otín et al., 2013*). In light of previous cross-sectional health studies in mice (*Justice et al., 2014*; *Fischer et al., 2016*), we performed an extensive characterization of the group differences in several relevant phenotypes, including glucoregulatory and physical performance metrics, in a cohort of male C57BL/6N mice ranging from 3 to 36 months of age. In addition, untargeted metabolite profiling of multiple organs and serum was conducted in this cross-sectional study, allowing a thorough characterization of the functional and molecular changes that occur with age. We report group differences in many functional and phenotypic biomarkers of aging in mice and provide an initial characterization of the specific biomolecular changes that occur in different organs that may contribute to these age-related conditions.

## Results

### Physiological, biochemical, and physical behavioral tests in a cross-sectional study of aging in mice

To characterize aging phenotypes in a cohort of male C57BL/6N mice, we utilized physiological, biochemical, and physical behavior tests to cross-sectionally assess group differences in these traits with age. Age group classification was as follows: Young (Y), (3–8 months; n = 23); Adult (A), (13–23 months; n = 26), and Old (O) (27–36 months; n = 22) (*Figure 1A*).

Frailty was the first characteristic assessed due to its high human relevance as an established aging biomarker and good predictor of clinical outcomes. It is characterized by loss of gait speed, fatigue, loss of body weight and muscle strength, and cognitive decline (*Palliyaguru et al., 2019*). Here, we used a multimodal approach to assess the impact of aging on the development of frailty and the associated impairment of several related phenotypic measures. The frailty index (FI) is a compilation of 31 clinical parameters, with 0 being not frail and one being frail (*Whitehead et al., 2014*). The values obtained from FI assessment increased progressively among different groups of mice, passing from $0.13 \pm 0.01$ in group Y to $0.24 \pm 0.01$ in group A, and $0.31 \pm 0.02$ in group O (*Figure 1B*), in agreement with earlier results (*Whitehead et al., 2014*). The FI increased linearly as a function of age (*Figure 1B*, p=3.80e-16, $r^2$ = 0.615), with age accounting for 61.5% of FI variation. These group differences in FI were associated with reduced forelimb muscle strength, expressed as latency to fall from a wire (p=0.0033, $r^2$ = 0.113), and gait speed (p=2.73e-16, $r^2$ = 0.608)

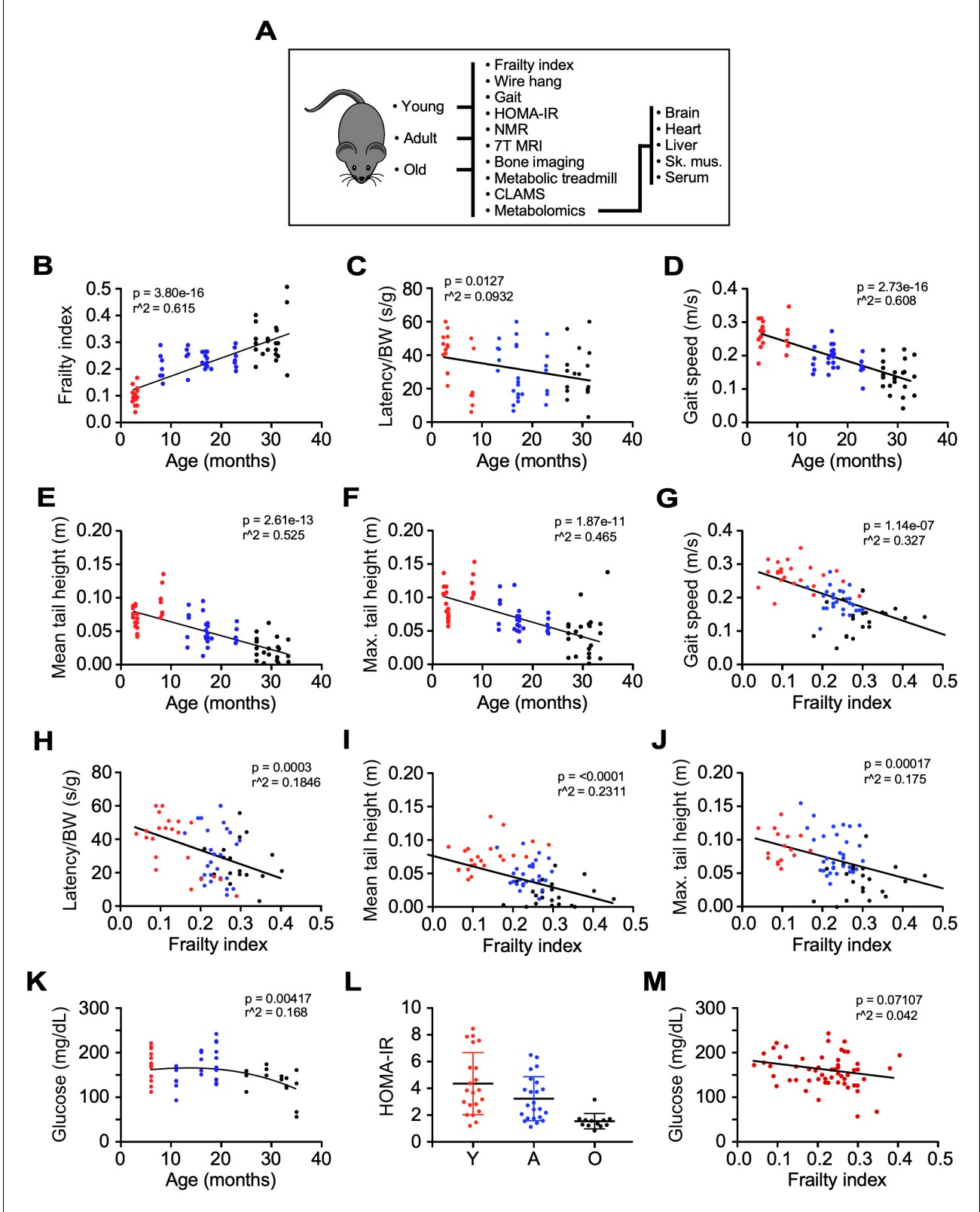

**Figure 1.** Impact of age and frailty on motor function, gait, and glucose homeostasis in male C57BL/6N mice. (A) Flowchart depicting analyses performed in the study. (B–D) Effects of age on frailty index, motor function, and gait speed in mice. (E–F) Mean and maximum tail height as a function of age. (G–J) Negative correlation between gait speed, motor function, and tail height with frailty. (K) Dependence of fasting blood glucose with age. (L) Scatter plot depicting insulin sensitivity as measured by the HOMA-IR index. (M) Relationship between fasting blood glucose and frailty index.

*Figure 1 continued on next page*

Figure 1 continued

Glucose measurement was performed 3 months after frailty assessment. (B–M): Mice ranged in age from 3 to 33 months. Y: young (3–8 mo, n = 23); A, adult (13–23 mo, n = 26); O: old (27–33 mo, n = 22). The actual number of data points shown on the graphs varies as not all mice were tested for any given intervention.

The online version of this article includes the following figure supplement(s) for figure 1:

**Figure supplement 1.** Phenotypic characteristics of male mice in function of age.

(*Figure 1C,D*). Age accounted for 52.5% and 46.5% of the mean (p=2.61e-13) and maximal (p=1.87e-11) tail height variation, respectively (*Figure 1E,F*). Given the strong relationship between FI vs. age and FI's significance as a metric of age-dependent decline, we related other functional parameters, for example, wire hang, tail height and gait speed, to FI. Gait speed exhibited the strongest relationship with FI from that of age itself, since FI can explain ~33% of the gait speed decrease (*Figure 1G*) compared to only ~17–25% of the other variables (*Figure 1H–J*). As expected, clear deficits in muscle strength (wire hang test and mean/maximal tail height) and walking speed were significantly related with FI. Interestingly, since total gait speed variation could, in principle, be explained by age and FI (*Figure 1B and G*, respectively), it can be considered a major biomarker of aging-dependent decline. After adjusting for age, frailty lost the association with gait speed (*Figure 1G*, p=0.477) but it remained strongly associated with forelimb muscle strength (*Figure 1H*, p=0.00088). After including age, FI was only marginally associated with mean tail height (*Figure 1I*, p=0.0792) while still being associated with maximal tail height (*Figure 1J*, p=0.0495). The effects of frailty on muscle strength were maintained after correction for age, indicating the presence of independent risk factors beyond chronological age that could reduce physical performance in the oldest mice.

Given the adaptive role of insulin resistance in aging (*Barzilai and Ferrucci, 2012*), venous blood was collected from 6 hr fasted mice. Blood glucose, plasma insulin concentrations, and HOMA-IR index, a surrogate for estimating insulin resistance (*Matthews et al., 1985*), were largely similar from 6 to 16 months and were observed to be lower after 19 months of age (*Figure 1K,L*, *Figure 1— figure supplement 1A*). In contrast to physical performance measurements, only a weak association existed between fasting blood glucose and the FI score (*Figure 1M*). These results may reflect a survivorship bias of the long-living mice (*Scheen, 2017*), although other effects such as reduced body weight and age-associated regional distribution among fat depots could have profound metabolic implications vis-à-vis changes observed in fasting blood glucose and insulin resistance (*Girard and Lafontan, 2008*; *Hardy et al., 2012*; *Tran et al., 2008*; *Shuster et al., 2012*).

In order to understand the impact of body composition on overall fitness, body composition analysis was performed next using low-field NMR spectroscopy (*Figure 2A*). Group differences in body weight and whole-body fat percentage were noted with a steady increase in values up to ~19 months and a decline thereafter (p=7.87e-07, $r^2$ = 0.319 and p=2.71e–07, $r^2$ = 0.340, respectively), whereas a weak positive association was observed between fasting glucose levels and percent body fat (*Figure 1—figure supplement 1B*). To determine body fat distribution, a 7T MRI body scanner was used to quantify the subcutaneous/visceral fat ratio, which revealed a linear reduction in visceral adiposity as a function of age (p=3.38e-05, $r^2$ = 0.2996) (*Figure 2B*). The whole-body lean tissue percentage was largely preserved during aging (p=2.15e-05, $r^2$ = 0.250) (*Figure 2A*). Representative MRI scans from groups Y, A, and O are depicted (*Figure 1—figure supplement 1C*).

Age-associated bone loss is reflective of a general functional decline. Old mice exhibited marked and significantly lower femoral cortical bone thickness (p=7.20e-07, $r^2$ = 0.458) and trabecular bone mineral density (BMD) (p=0.0184, $r^2$ = 0.112) compared to the other two groups of mice (*Figure 2C*). Representative micro-CT scans of the femoral region are depicted (*Figure 1—figure supplement 1D*). No association between BMD and body weight was found (*Figure 1—figure supplement 1E*). Remarkably, there was a lower percentage of fat around the tibia per unit of body mass in animals of increased age, with the oldest group of mice being the most affected (p=0.002, $r^2$ = 0.248), whereas the percentage of lean tissue surrounding the tibia trended higher in group O compared to group A (*Figure 2C*). The presence of weaker bones is suggestive of age-related osteoporosis.

To assess whether the lower gait speed in old animals was caused by an altered cardio-metabolic phenotype, mice walked on a metabolic treadmill at their age-group's average gait speed

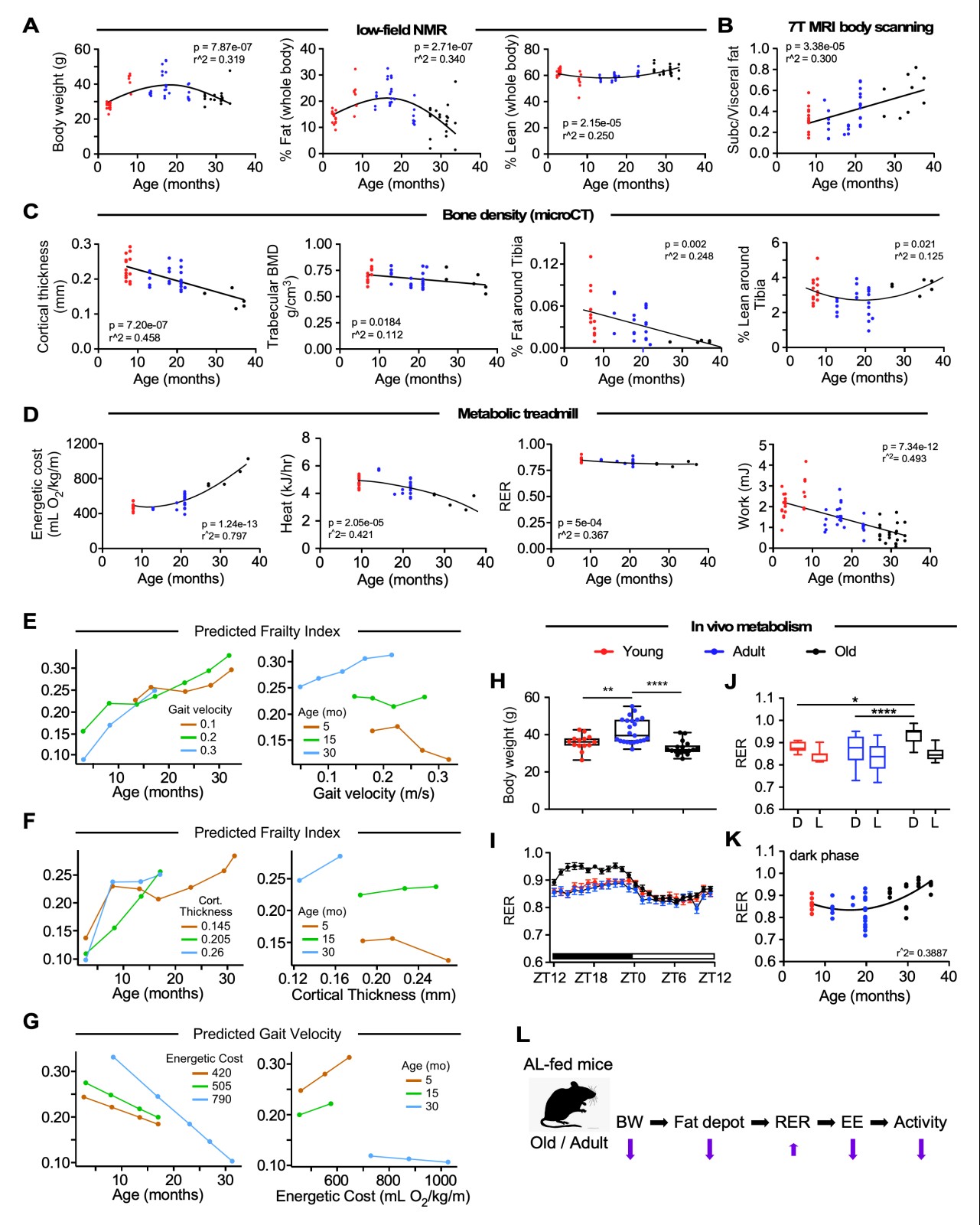

**Figure 2.** Impact of age on body composition measurements, bone density, and in vivo metabolic function in male C57BL/6N mice. (**A**) Body weight (left panel), percentages of whole-body fat mass (middle panel) and lean tissue mass (right panel) as assessed by low-field nuclear magnetic resonance imaging. (**B**) Ratio of subcutaneous to visceral fat calculated from abdominal 7-Tesla MRI scanning. (**C**) In vivo microCT analysis of the tibia was carried out to measure cortical thickness, trabecular bone mineral density (BMD), as well as fat and lean mass around the tibia expressed as percentage of the

*Figure 2 continued on next page*

*Figure 2 continued*

total body weight. (D) Metabolic parameters derived from the metabolic treadmill as a function of age. Energetic cost (left panel); heat generation (second panel); RER, respiratory exchange ratio (third panel); and Work (right panel). Energetic cost was determined from $VO_2$ per unit of body weight and gait speed while work was expressed as mJoules. (E–G) Multiple regression analyses with frailty index (E, F) and gait velocity (G) as response variables. The modeling approach consisted of fitting a series of four regressions for each of the response variables. The explanatory variables included in the multiple regression modeling are depicted in *Figure 2—source data 1*. (E) *left panel*, Effect of gait velocity on the relationship between frailty and age; *right panel*, Impact of age on the interaction between frailty and gait velocity. (F) *left panel*, Effect of bone cortical thickness on the relationship between frailty and age; *right panel*, Impact of age on the interaction between frailty and bone cortical thickness. (G) *left panel*, Effect of energetic costs on the relationship between gait velocity and age; *right panel*, Impact of age on the interaction between gait velocity and energetic costs. Although this is a cross-sectional study, the points were joined to help the reader see the various associations without suggesting that there is causal association; for example, that animals with higher age tend to have higher frailty within the same gait velocity (E, *left panel*). (H–K) Mice at different ages were placed into metabolic cages for 72 hr to measure $VO_2$, $VCO_2$, RER, heat production, and ambulatory activity counts. The values associated with the first 12 hr acclimatization phase (L1) were discarded. (H) Body weight as mice entered the metabolic cage. (I) Averaged hourly RER trajectories were captured during two dark/light cycles in young (n = 8), adult (n = 22) and old (n = 18) mice. ZT, zeitgeber time is defined as the 12:12 hr light/dark cycle that synchronizes organismal biological rhythms. Each point represents mean ± SEM. (J) Scatter plot depicting RER values averaged over 48 hr for individual mice in the three age groups. (K) Relationship between RER and age. Data in (H) and (J) were analyzed using one-way ANOVA with Dunnett's post-hoc analysis. *, $p < 0.05$; **, $p < 0.01$; ****, $p < 0.0001$. (L) Diagram depicting the impact of age on body composition and energetic parameters in AL-fed male mice.

The online version of this article includes the following source data and figure supplement(s) for figure 2:

**Source data 1.** Multiple regression analyses results for frailty index and gait velocity.

**Source data 2.** Area under the curves (AUCs) from the averaged, hourly trajectories of various energetic parameters that were captured during two dark/light cycles in young, adult, and old mice.

**Source data 3.** Distribution of various energetic parameters in male mice in various age groups.

**Source data 4.** Linear regression of the metabolic cage data obtained from the whole cohort of male mice and after segregation by age groups.

**Source data 5.** Generalized linear model (GLM) to assess potential significant interaction effect in energetic parameters between mice of various age groups.

**Figure supplement 1.** In vivo metabolic data by indirect calorimetry.

(determined from the MotoRater data, see *Figure 1D*), while $VO_2$ and $VCO_2$ were simultaneously monitored. Although variability exists within age groups, there is a steady decline in gait speed measured from progressively older cohorts of mice, and therefore average gait speed emulated the natural walking pattern, which is defined as walking without hesitation (e.g. stopping frequently/ exploring/sniffing) not sprinting across the platform. Remarkably, the energetic cost of walking increased exponentially from ~20 months of age and was paralleled by a sustained decrease in work capacity (*Figure 2D*, 1st and 4th panels). The running $VO_2$ values did not differ over the entire activity episode (800 s.) across age, thus ruling out a role for fatigue in the higher energetic cost in the oldest mice (data not shown). Respiratory exchange ratio (RER, defined as $VCO_2/VO_2$) varies between ~0.7 (i.e. predominant use of fats as fuel source) and 1.0 (i.e. exclusive carbohydrate utilization), with intermediate RER values implying mixed substrate use. Interestingly, RER and heat dissipation values, obtained during the treadmill monitoring, decreased from ~0.85 to ~0.80, and from 5 to 3 kJ h$^{-1}$, respectively, indicating higher utilization of lipid than glucose as an energy source, and less heat dissipation in the older groups of animals (*Figure 2D*, 3rd and 2nd panels). The differences in work economy and heat production between the three age groups were maintained after adjusting for lean mass (data not shown). The diminished heat dissipation *vs.* age might not be due to an increase in the efficiency of energy use by the group O, but rather a predominant overall decrease in work capacity, that is, by working less, the mice dissipate less heat (*Figure 2D*, 2nd and 4th panels). These results are also consistent with the observation that, compared with young animals, older mice consume significantly more $O_2$ (*Figure 2D*, 1st panel) and released more $CO_2$ per unit gait speed (not shown), because enhanced lipid catabolism demands more $O_2$.

Multiple linear regression analysis showed that from the initial set of explanatory variables, the interaction of gait velocity and age was found to be significant (*Figure 2E*, left panel). Frailty status was more apparent with age as gait velocity increases: Specifically, at younger ages, the frailty index was inversely correlated with gait velocity while at older ages frailty status showed strong association with lower gait velocity (*Figure 2E*, right panel). Studies have shown that higher level of *HOMA*-IR is significantly associated with frailty in the elderly, but such association is not observed in the middle-aged population (*Peng et al., 2019*); moreover, fasting plasma glucose does not differ between frail

and non-frail individuals (*Kalyani et al., 2012*). Here, positive associations between HOMA-IR tertiles (Q1, median, Q3) and the risk of frailty were observed in old animals, while fasting blood glucose tertiles and frailty status were negatively correlated (*Figure 2—source data 1*). By incorporating bone density variables into the starting model, we found that frailty increases more steeply with age as cortical thickness increases (*Figure 2F*, left panel) and that the effect of cortical thickness on frailty index is much larger in older ages (*Figure 2F*, right panel). The introduction of variables such as percent lean mass, percent fat and visceral-to-subcutaneous fat ratio as well as metabolic treadmill parameters such as energetic cost, heat generation or RER did not improve the initial model (*Figure 2—source data 1*).

Of all the experimental variables analyzed, age and energetic costs were the only explanatory variables for gait speed (*Figure 2—source data 1*). Gait speed tended to be lower in older animals than in group Y, consistent with the observations made in the human population (*Studenski et al., 2011*). Moreover, at higher energetic costs there was a stronger decreasing relationship between gait velocity and age (*Figure 2G*, left panel). There was a marked positive association between gait velocity and energetic costs in groups Y and A, whereas no significant relationship was observed in group O (*Figure 2G*, right panel).

To better understand the impact of age on energy balance and availability in mice, we conducted indirect calorimetry experiments in metabolic chambers for 72 hr to collect specific measurements such as the $VCO_2$, $VO_2$, RER, energy expenditure (EE), and ambulatory activity counts. Mice from Group A (12–20 mo) were significantly heavier than group Y (7 mo) and group O (26–36 mo) animals as they entered the metabolic cages (*Figure 2H*). Hourly trajectories of energetic parameters were captured over two 24 hr cycles (D1-L2-D2-L3) and then were averaged. Distinctive diurnal patterns of $O_2$ consumption and $CO_2$ production were observed in the three groups of mice (*Figure 2—figure supplement 1A*) that led to differences in the calculated values for RER (*Figure 2I*), with significantly higher area under the curve (AUC) in group O vs. groups Y and A, indicative of greater metabolic flexibility (AUC for Y, 2.563 ± 0.1163 [CI 2.335–2.791]; A, 2.311 ± 0.2188 [CI 1.883–2.74]; O, 3.277 ± 0.1421 [CI 2.999–3.556]). Moreover, EE was greater in group A compared with the other two groups consistent with their higher $O_2$ consumption (*Figure 2—figure supplement 1A*). While these differences are largely reflective of the higher body mass of group A mice, it was also clear that the relationships between these metabolic parameters and body weight were altered with age (*Figure 2—source data 2*). Lastly, the higher ambulatory activity during the dark cycle was significantly lower in group O mice (*Figure 2—figure supplement 1A*, *Figure 2—source data 2*) and showed a strong negative correlation with age across the entire pool of animals. The diurnal changes in RER in the three age groups showed statistically significant differences in the dark cycle (D) between group Y vs. group O and group A vs. group O (*Figure 2J*), with higher RER in the oldest group of mice (*Figure 2K*). No difference in RER could be observed in the light phase between the different groups of mice. Scatter plots of the averaged 48 hr values of various energetic parameters from groups Y (n = 8), A (n = 22), and O (n = 18) mice showed statistically significant differences in group Y vs. group A mice and A vs. O mice in $VO_2$ and EE (*Figure 2—figure supplement 1B*, *Figure 2—source data 3*). In contrast, there were no differences between groups Y and O mice in any of the measures except for the ambulatory activity (*Figure 2—figure supplement 1G*, *Figure 2—source data 3*).

Consistent with the lower gait speed observed in older animals, metabolic cage analysis reveals a strong negative correlation between age and ambulatory activity (*Figure 2—figure supplement 1C*), whereas no other respiratory parameters correlated significantly with age (*Figure 2—source data 4*). As expected, body weight was positively correlated with EE, oxygen consumption, and $CO_2$ production (*Figure 2—figure supplement 1D*, *Figure 2—source data 4*). The observed differences in the AUC of respiratory parameters were primarily a result of the higher body weight of group A mice, which is why a generalized linear model (ANCOVA) analysis was used to determine differences in the relationship of body weight with $VO_2$, $VCO_2$, and EE (*Figure 2—figure supplement 1E*, *Figure 2—source data 5*). This analysis demonstrated statistically significant differences in the relationship between body weight and respiratory parameters when comparing group Y vs. A mice and group Y vs. O mice. These metabolic alterations may complement many of the differences that were observed in body composition and other related parameters.

Overall, the data presented in this section show that gait speed is a major biomarker of healthspan, and that frailty, in addition to age, is able to explain to a great extent its functional

deterioration. In contrast, morphometrics (e.g. body weight, % fat and % lean mass) and glucose homeostasis (e.g. blood glucose, insulin, and HOMA2-IR) metrics show a more nuanced relationship with age. In fact, many of these factors (e.g. blood glucose, insulin, body weight, % fat) do not start to show a reduction in values until after ~19 mo, whereas others (e.g. bone density parameters, % fat around tibia) steadily decline in the values from various groups of mice with age. Important exceptions to this include % lean mass, % lean mass around the tibia per whole body weight, and the ratio of subcutaneous/visceral fat, that tend to be higher in group O animals. These findings provide a coherent picture of the bioenergetic changes that occur as a function of age, marked by an exponential increase in the energetic cost of physical activity, which is paralleled by a sustained decrease in work (energy). Together with gait speed, these parameters constitute landmarks of aging-dependent decline (*Figure 2L*), which led us to explore in further detail how the metabolomic landscape was altered with age in different tissues.

## Untargeted metabolomics analysis

To determine whether the differences in healthspan assessments, such as physical function, FI, energetics, and work economy can be linked to metabolic remodeling, we performed untargeted metabolomics analysis. We used serum and four different organs (brain, heart, liver, and skeletal muscle) of the same mice and compared the three different life stages: Y, A, and O. Depending upon the organ, a total of 133 to 177 metabolites was detected. Combining univariate and multivariate statistics we identified 49 metabolites that were significantly different in serum and across organs as a function of age, of which 46 were shared. *Figure 3* depicts a heat map of the averaged levels of the 46 shared metabolites after normalization as a function of age. The metabolite abundance in each mouse is provided (*Figure 3—figure supplement 1*).

Several features are worth highlighting: (i) The liver was the organ exhibiting the highest abundance of metabolite intermediates from all major pathways involved in macronutrient metabolism (glucose, lipids, amino acids) and redox-related pathways (nicotinamide, glutathione, pentose phosphate); (ii) heart and serum had the next highest metabolite abundance; (iii) across age and organ, there was an apparent qualitative similarity of the metabolites' pattern of abundance or depletion; however, significant quantitative differences in metabolite levels also exist, as revealed by multivariate statistics (see *Figure 4*); (iv) several metabolites from redox-related pathways, such as pentose-phosphate, $NAD^+$ salvage, methionine cycle, and transsulfuration, exhibited significant differences with age; and (v) expectedly, relative accumulation of glutamate/glutamine in the brain, and of lactate in serum was observed – the latter concomitant with depletion in the liver where it is utilized for gluconeogenesis.

Applying principal component analysis (PCA), a multivariate clustering method, to the metabolomic dataset, we found a clear separation between the metabolite profiles among organs and serum for each age group (*Figure 4A*).

## Adaptive redox-related metabolic remodeling is a common trait in old mice

We next determined if groups of metabolites were different in each organ in Y, A, and O mice. As revealed by partial least squares discriminant analysis (PLSDA), a supervised clustering method, brain and heart exhibited a net separation between the three groups of mice. In contrast, in liver, skeletal muscle and serum, Y and O animals were clearly separated while group A mice partially overlapped with both groups (*Figure 4B*). To identify the main metabolites responsible for age-dependent metabolic differences, we analyzed the top 20–25 metabolites in each organ, ranked based on the variable importance in projection (VIP) score as a function of age (*Figure 4C*); the mini heat map on the right shows each metabolite's level variation within the different age groups.

The age dependence of each metabolite's concentration was further assessed by quantifying the Pearson correlation coefficient as a function of age; positive and negative values denote greater or lower levels in older mice. Depicted in *Figure 4D* are the brain, heart, liver, serum and skeletal muscle results. For example, the top sixteen VIP score-ranked metabolites (VIP >1.0) in the liver are also among the most important ones increasing (F6P, ribulose 5P, G6P, glutathione) or decreasing (pyruvate, xylitol, sorbitol, nicotinamide) with age. Extending this analysis to the other organs enabled us to visualize the emergence of a consistent 'meta-pattern' of metabolic change characterized by

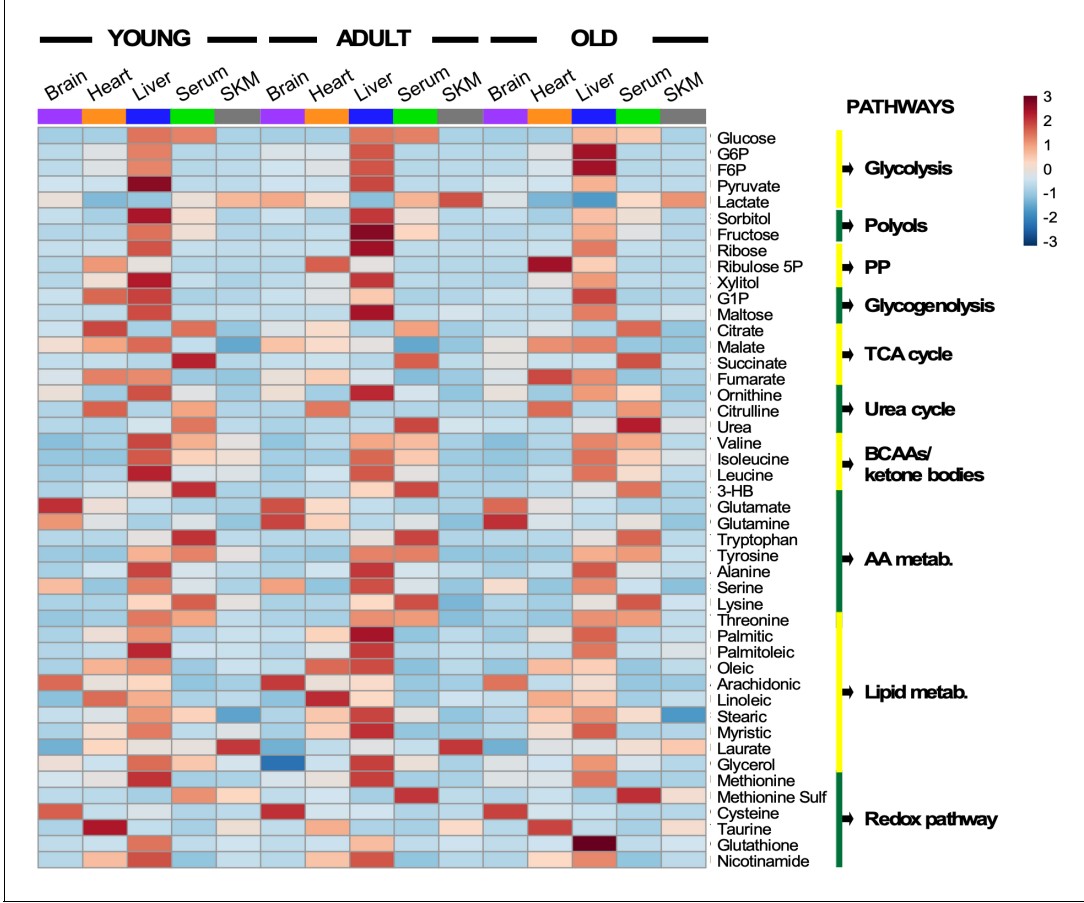

**Figure 3.** Heatmap depicting levels of modulated metabolites and associated pathways in brain, heart, liver, serum, and skeletal muscle from young, middle-age, and old mice. There were 46 metabolites involved in more than 10 defined pathways that were shared between serum and various organs across the three age groups. Each row represents a single metabolite, and each column depicts averaged values for each tissue and age group (In each tissue, n = 11–12 animals for young; n = 7 for adult; and n = 11–12 for old). For each metabolite, standardized abundance was calculated. The pseudocolor scaling of the standardized expression is from low (blue) to high (red).

The online version of this article includes the following figure supplement(s) for figure 3:

**Figure supplement 1.** Heatmap depicting levels of regulated metabolites and associated pathways in brain, heart, liver, serum, and skeletal muscle from young, adult, and old mice.

metabolites from redox-related pathways, such as pentose phosphate (G6P, ribulose 5P, ribose 5P), NAD$^+$ salvage (nicotinamide), methionine cycle (methionine) and transsulfuration (cysteine, taurine, glutamate, glycine, glutathione) (*Figure 4D*). Unlike in liver and the heart, where higher relative accumulation levels of those metabolites were detected in older animals, skeletal muscle exhibited opposite comportment whereas the brain displayed an intermediate behavior. Specifically, G6P, ribulose 5P, ribose 5P and glutathione were elevated with age in liver and/or heart (*Figure 4D*, *Figure 4—figure supplement 1*). There was also an apparent difference in the response of these organs to oxidative stress, as revealed by lower nicotinamide and cysteine in the liver, and higher methionine sulfoxide (a biomarker of oxidative stress that originates from oxidation of methionine) in the heart from aged mice. In skeletal muscle, nicotinamide, methionine and cysteine were lower in group O mice, whereas in the brain, nicotinamide was relatively low, in contrast to methionine, which was modestly higher, while methionine sulfoxide was diminished (*Figure 4D*, *Figure 4—figure supplement 1*). Interestingly, the higher levels of methionine sulfoxide in the serum from old animals along with lower levels of nicotinamide are consistent with increased oxidative stress as a function of age (*Figure 4D*).

Together, these data suggest that as mice age, organs such as liver and heart, which are exposed to higher oxidative stress as a result of their function (detoxification in the liver and energy

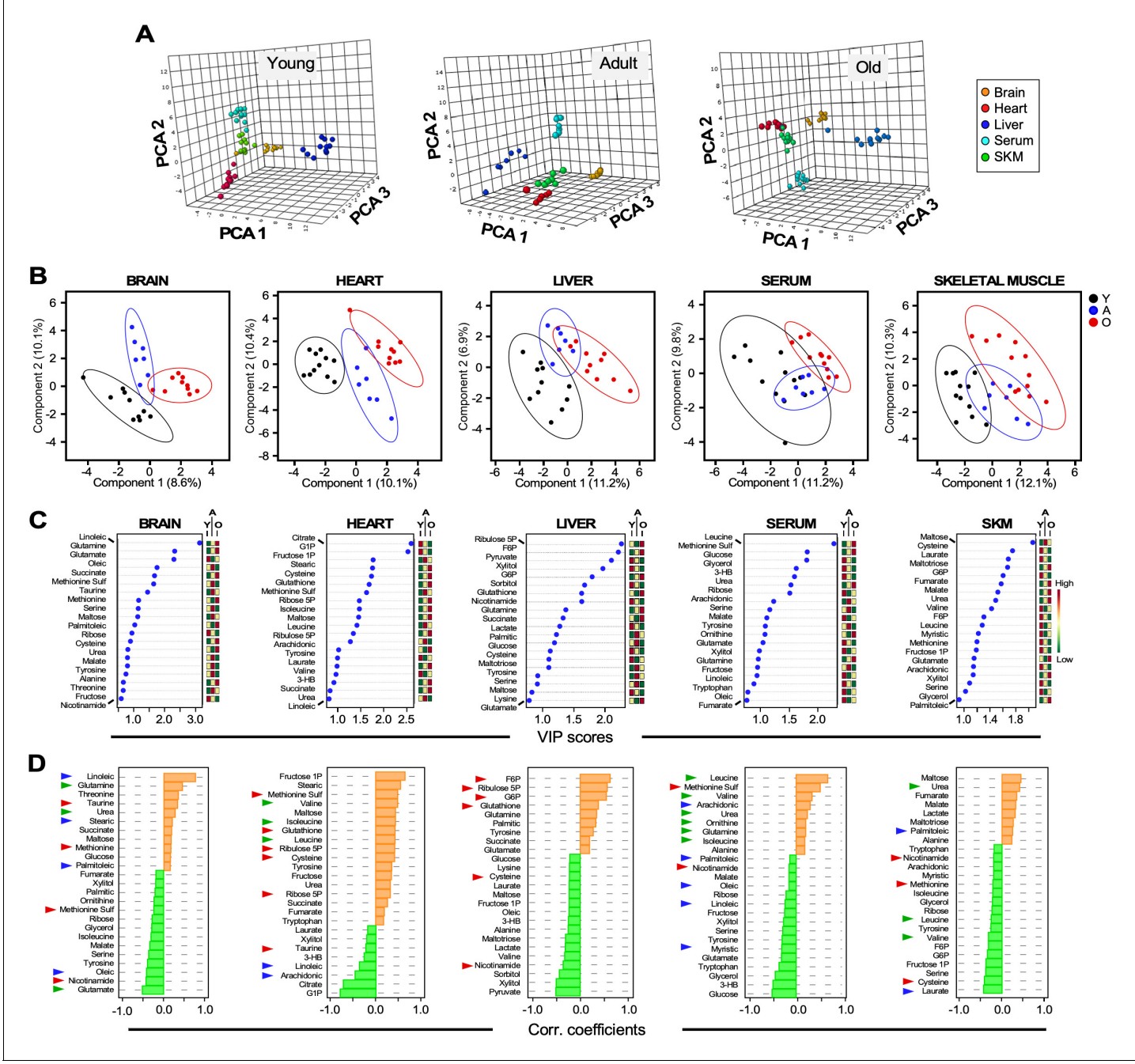

**Figure 4.** Untargeted metabolomics performed in multiple organs and serum from young, adult, and old male C57BL/6N mice. (**A**) Three-dimensional principal component analysis (PCA) depicting the effect of age across various organs and serum. (**B**) Brain, heart, liver, serum, and skeletal muscle metabolite profiles from young (Y, black symbols), adult (A, blue symbols) or old (O, red symbols) mice were analyzed by Partial Least Square Determinant Analysis (PLS-DA). A statistically significant degree of separation was observed between age groups. The ellipses correspond to 95% confidence intervals for a normal distribution. Each principal component is labeled with the corresponding percent values. (**C**) Variable in projection (VIP) scores of positively (red boxes) and negatively (green boxes) correlating metabolites in each tissue or serum as a function of age. (**D**) Correlation coefficients of the top 25 metabolites that correlated positively (orange bars) and negatively (green bars) as a function of age for the indicated tissues and serum. Blue arrowhead, fatty acids; green arrowhead, amino acids and related; red arrowhead, redox and related.

The online version of this article includes the following figure supplement(s) for figure 4:

**Figure supplement 1.** Heatmaps depicting average values of the metabolites enriched in brain, heart, liver, serum and skeletal muscle, respectively, as a function of age.

generation in the heart), remodel their metabolism toward higher expression/activity of redox-related metabolic pathways, for example, pentose phosphate, NAD$^+$ salvage and transsulfuration. Oppositely, skeletal muscle and brain do not appear to be able to remodel those pathways thus becoming more vulnerable to increased oxidative stress with aging. This metabolic pattern agrees with the idea of different rates of aging among organs.

## Age and organ-specific metabolic remodeling

Next, we addressed the metabolic pathways involved in this remodeling, according to metabolite intermediates present in the profile. Serum metabolite profiles from old mice showed accumulation of intermediates from urea cycle function (urea, glutamine, ornithine), a pattern consistent with pre-dominant amino acid degradation in the liver, and significant accumulation of the branched-chain amino acids (BCAAs) valine, isoleucine and leucine (*Figure 4D*, *Figure 4—figure supplement 1*). Remarkably, the three top-ranked metabolites, glucose, 3-hydroxybutyrate (3-HB, a ketone body), and glycerol appear to be lower in the serum of old mice, suggesting a reduced supply of these fuel substrates from liver and adipose tissue to heart, brain, and skeletal muscle (*Figure 4D*, *Figure 4—figure supplement 1*). Together, the lower levels of glycerol and serum fatty acids (myristic, linoleic, oleic, and palmitoleic, but not arachidonic) suggests potential liver and adipose tissue dysfunction in aged mice, in agreement with the body composition data and the metabolic cage data.

In liver, depletion of glucose, pyruvate, sorbitol, fructose 1P (F1P), xylitol, maltose, and malto-triose with age suggests active glycolysis and glycogen breakdown, less utilization of the polyol pathway, and hepatic release to the circulation (*Figure 4D*, *Figure 4—figure supplement 1*). The accumulation of tyrosine, tryptophan, glutamine, and glutamate, concomitant with the depletion of alanine and valine (but not glycine), is mirrored by the serum metabolite profile, consistent with active amino acid degradation and urea cycle function.

The heart shows a remarkable accrual of BCAAs with age. Glucose degradation pathways appear to be all very active including polyols (fructose, F1P) and glycogenolysis (maltose) also consistent with glucose 1P (G1P) depletion (*Figure 4D*, *Figure 4—figure supplement 1*). Higher levels of the tricarboxylic acid intermediates fumarate, malate, and succinate suggest active mitochondrial function. Reduced production of 3-HB further contributes to the loss of 3HB circulating levels and reveals less fatty acid oxidation. In the brain, the accumulation of the top-ranked fatty acid ω6 linoleic acid together with stearic and palmitoleic acids is notably associated with the increased abundance of 3-HB, a preferred substrate of this organ (*Figure 4D*, *Figure 4—figure supplement 1*). The higher levels of urea and glutamine in older mice, accompanied by depletion of ornithine, suggest active urea cycle function, a trait consistent with depletion of several amino acids (isoleucine, serine, tyrosine, glutamate) (*Figure 4D*, *Figure 4—figure supplement 1*).

Expectedly, the skeletal muscle from old mice exhibits high glucose catabolism as suggested by the accumulation of maltose, lactate and maltotriose. Higher levels of urea concomitant with lower levels of several amino acids, such as leucine, tyrosine, valine, serine, and cysteine, indicate of active amino acid degradation accompanied by urea cycle function (*Figure 4D*, *Figure 4—figure supplement 1*).

Metabolite set enrichment analysis (MSEA) shows the statistical significance of the metabolic pathways involved for each organ. The results obtained support the idea that metabolic remodeling in old mice occurs via redox-related pathways in liver and heart as compared to skeletal muscle and brain (*Figure 5*). MSEA also confirmed the relevance of the urea cycle and ammonia cycling likely due to amino acid degradation in skeletal muscle, brain, and liver, a metabolic trend that is also reflected by the serum metabolite profile. With aging, and in an amphibolic organ like the liver, amino acid degradation could be triggered in parallel with protein biosynthesis, as indicated by tissue enrichment (*Figure 5*). In the aging phenotype of predominantly postmitotic catabolic organs like skeletal muscle, heart, and brain, protein synthesis might be less significant compared to that in liver.

Together, the data obtained show that each organ exhibits an age-specific metabolic remodeling which is, in part, associated with its function, for example, prominence of redox-related pathways in liver and heart, active glucose catabolism in skeletal muscle, activation of glucose metabolism in the heart and gluconeogenesis and urea cycle function in liver. Statistically significant pathways found by MSEA in old mice are consistent with major hallmarks of metabolic remodeling identified by other types of analysis. Many of the differences observed in this metabolomic profiling can be linked to

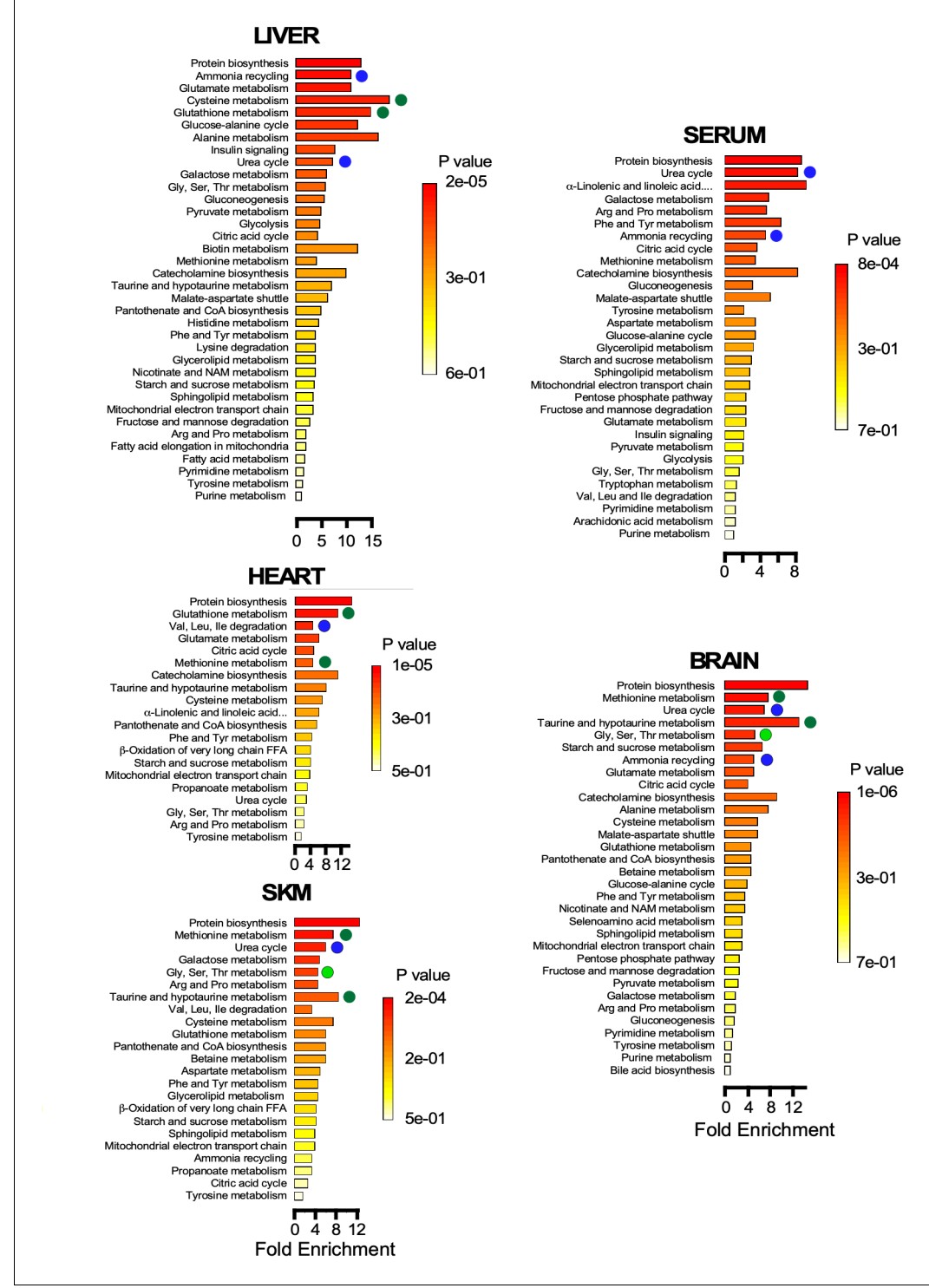

**Figure 5.** Pathway enrichment in brain, heart, liver, serum, and skeletal muscle Metabolite set enrichment analysis (MSEA) showing the statistical significance of the metabolic pathways involved for each organ. Note the metabolic remodeling occurring with age, as it pertains to redox-related pathways (green symbols), urea cycle and ammonia cycling through amino acid degradation (dark blue symbols), and one-carbon metabolism (light green symbols).

the phenotypic alterations that were observed in group O mice, either as factors that may be involved in promoting or may arise as a result of these differences.

## Patterns of metabolic signatures of biological aging from covariation of metabolites with physiological metrics

In order to cope with the increased energetic cost of maintaining health and energy balance, each organ must fine-tune their metabolism and substrate selection. To assess the strength of the links between metabolites and phenotypic outcomes, we generated correlation matrices between energetic cost, RER, or frailty index (FI) and metabolites representative of major pathways involved in the metabolic remodeling of the four organs (heart, skeletal muscle, liver, and brain) and serum. Firstly, a correlation matrix with the 46 significantly changed metabolites was constructed to investigate their covariation, independently of age and organ (*Figure 6A*). A big cluster of positively correlated metabolites accompanied by four minor ones were found, from which 24 metabolites from all clusters and representative pathways were selected. The energetic cost of walking and RER are two metrics closely related to gait speed, whereas FI, a major biomarker of aging, is only remotely related to metabolism. Next, the Pearson correlation coefficient for each of the 24 metabolites was calculated with respect to the energetic cost, RER or FI from scatter plot matrices ('r' with $p < 0.05$) for each of the four organs and serum (see *Figure 6—figure supplement 1* as an example and *Figure 6—source data 1*). Hierarchically clustered heatmaps allowed for visualization of all correlation coefficients (positive and negative) corresponding to the 24 metabolites as a function of the phenotypic outcome in each organ and serum (*Figure 6B–D*). Broadly speaking, the increase in energetic cost, RER, or FI was correlated with organ-specific patterns of increased abundance or depletion of select metabolites.

With respect to the energetic cost of walking (*Figure 6B*), a group of metabolites show a strong negative correlation in liver, of which only pyruvate, palmitoleic acid, and glucose were shared by the heart. A second cluster of negatively correlated metabolites represented by nicotinamide, alanine, and myristic acid was present in heart, and to a lesser extent in liver and brain. BCAAs (valine, isoleucine, leucine) and several other metabolites, including 3-hydroxybutyrate (3-HB), correlated positively with energetic cost in the heart, while being negatively linked in liver. On the other hand, glutathione correlated positively with energetic cost in both organs (heart > liver). For many metabolites, including glutathione, glycerol, and methionine sulfoxide, the brain presented a distinct pattern with respect to the other organs. The skeletal muscle resembled the heart profile with some exceptions.

Taken as a whole, the increase in energetic cost correlates with a pattern of metabolites related to substrate utilization. This pattern is consistent with a mixture of glucose, lipid, and amino acid metabolism in liver, and more predominant glucose and lipid metabolism in the heart, whereas lipids appeared to prevail over glucose in skeletal muscle. Remarkably, brain displayed a pronounced amino acid catabolism in addition to the expected dependence on glucose and ketone bodies. Regarding redox metabolism, heart exhibited the highest correlation between energetic cost and glutathione, followed by liver then SKM, while ribose from the pentose phosphate pathway was positively correlated in heart only, and the oxidative stress biomarker methionine sulfoxide showed a positive correlation in both cardiac and SKM, suggesting a more oxidative environment in these two organs along with brain.

As a function of RER (*Figure 6C*), liver and heart shared a cluster of both negatively and positively correlated metabolites. Again, the brain exhibited a clearly distinct pattern with respect to the other organs consisting of mostly positively correlated metabolites, whereas skeletal muscle presented a more mixed pattern with two noticeable clusters of negatively and positively correlated metabolites. An elevated RER is indicative of an increase in the proportion of carbohydrates utilized for energy production. Consistent with this, we observed an increasing abundance of glucose intermediates in both liver and heart concomitant with depletion of lipids and amino acids. In this regard, SKM and brain presented a distinct pattern which was more based on glucose rather than lipid metabolism. The increasingly mixed glucose and lipid utilization led to a trend toward higher abundance of methionine sulfoxide and depletion in glutathione in response to increased RER in heart, liver, and SKM, while brain had markedly less methionine sulfoxide, but greater accumulation of methionine and nicotinamide compared to glutathione. This pattern is consistent with systemic oxidative stress and lower antioxidant capacity.

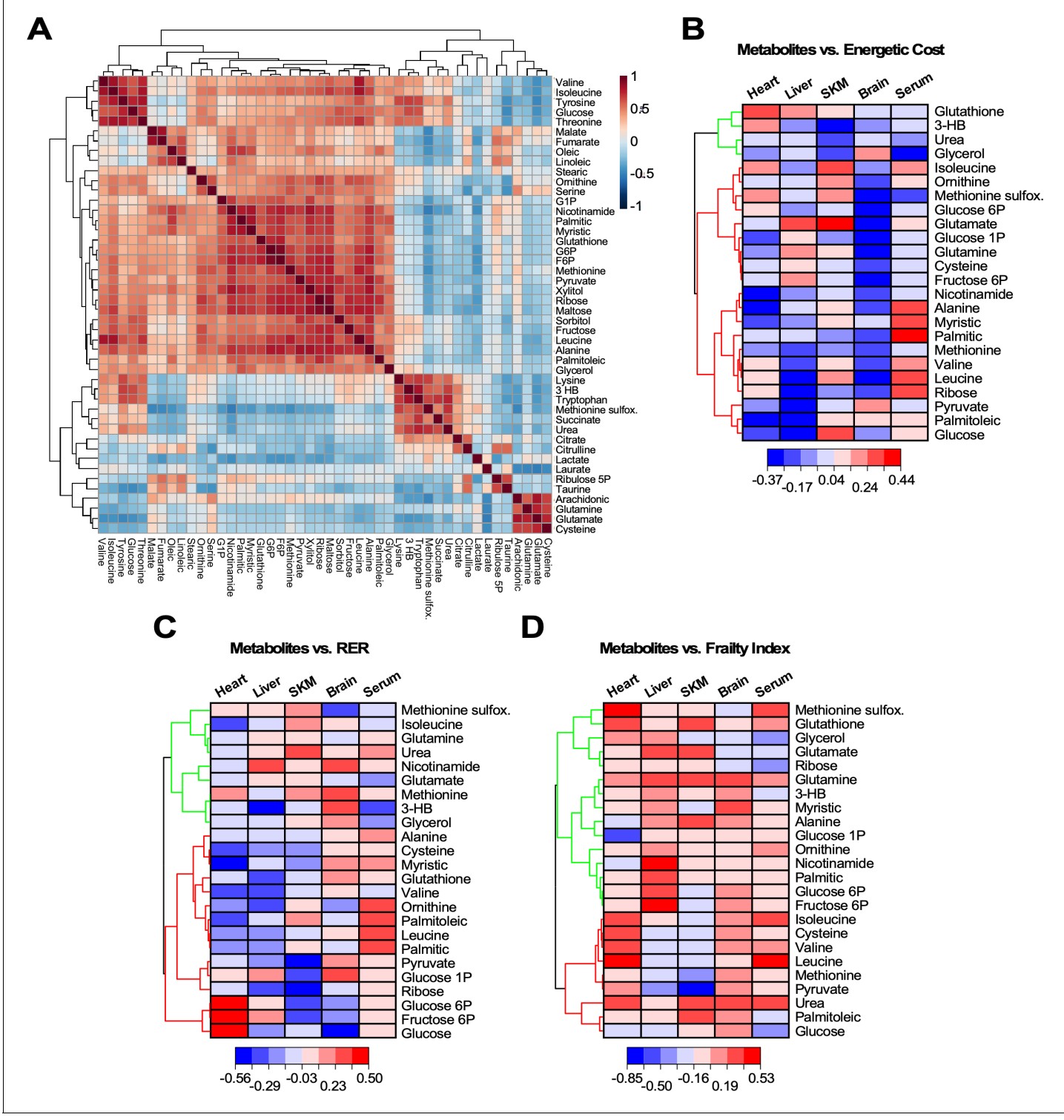

**Figure 6.** Heatmaps of hierarchically clustered correlation coefficients of representative metabolites from organs and serum vs. different physiological metrics. (A) Correlation matrix of the 46 metabolites significantly changed across age, organs and serum. Using the statistical module of MetaboAnalyst, we determined the clusters of normalized metabolites covarying positively or negatively, independently of organ or serum and age. The type (positive or negative) and strength (color intensity) of correlation are coded brown and blue, respectively, normalized between 1 and −1 according to the bar on the right. (B–D) Twenty-four significantly changed metabolites representative from all pathways that covary independently from organs or serum and age (see panel A) were correlated with energetic cost (B), RER (C) or frailty index (FI) (D). Using the software Origin v. 2021, we performed scatter plot correlation matrices (see *Figure 6—figure supplement 1*) to determine the Pearson correlation coefficient, r (p<0.05), for each
*Figure 6 continued on next page*

*Figure 6 continued*

of the 24 metabolites in each organ and serum vs. each of the physiological metrics analyzed. Displayed are in the form of hierarchically clustered heatmaps (dendogram on the left) corresponding to the ensemble of correlation coefficients, positive (red) and negative (blue), exhibited by each metabolite vs. the specified physiological metric.

The online version of this article includes the following source data and figure supplement(s) for figure 6:

**Source data 1.** The Pearson correlation coefficient, r, value for each metabolite vs. energetic cost, RER, and Frailty index.
**Figure supplement 1.** Scatter plot matrix of 14 metabolites vs. energetic cost (1st column) and of each metabolite with respect to each one of the other metabolites.

Increasing energetic cost and higher RER elicited an enrichment of several lipids and amino acids, but depletion of glycerol in serum (*Figure 6B and C*). These patterns are consistent with the metabolic remodeling exhibited by liver, a major sink of lipids and amino acids, and unveiled a reduced supply of 3-HB and glycerol from liver and adipose tissue to heart, brain, and skeletal muscle, at least in old mice.

The prevalent pattern of metabolites correlating positively with increasing FI (*Figure 6D*) appears to be inconsistent with the pattern profiles revealed by energetic cost and RER. This is expected given the less direct mechanistic relationship between metabolism and FI since the latter derives from a compilation of many different phenotypic features (*Palliyaguru et al., 2019*). Energetic cost results from the ratio of respiration ($VO_2$) over gait speed (*Figure 2D*), while RER includes only respiration-related variables ($CO_2$ release over $O_2$ uptake). Hence, because the latter is more directly related to metabolism, the mechanistic links between RER and metabolism can be more directly envisaged. Nevertheless, the energetic cost of walking/running is a relevant metric because of the presence of gait speed, a major biomarker of functional performance during aging. Remarkably, energetic cost and RER rendered a similar pattern of metabolic profiles (*Figure 6B and C*).

## Discussion

In this cross-sectional study, male mice of various ages (3–36 months) were assessed for functional differences associated with aging, while pursuing the detection of major biomarkers and underlying energetic and metabolic mechanisms. Metabolic remodeling in serum, liver, brain, cardiac, and skeletal muscle, their interactions and consequences for diminishing bodily functions, morphometry and biochemistry, were extensively tested and analyzed. We performed an in-depth characterization of the phenotypic differences that exist between groups Y, A, and O animals. This analysis demonstrated that (i) gait speed decline, which appears to be fully explained by age and frailty, is a major functional biomarker of aging in mice; (ii) the deterioration of locomotor activity is associated with a dramatic increase in the energetic cost of physical activity from age ~19 months, and was paralleled by a sustained decrease of working capacity; (iii) other morphometric (e.g. body weight, % fat and lean) and biochemical (e.g. blood glucose, insulin, HOMA-IR) metrics show a more subtle relationship with age. To further explore the molecular changes that may contribute to or result from these alterations, we performed metabolomic analysis in a number of key metabolic tissues and showed that (iv) different organs remodel their metabolism in response to specific functional demands, for example, energy supply and detoxification; specifically, unlike brain and skeletal muscle, liver and heart in old mice rewire metabolism toward higher expression/activity of redox-related metabolic pathways such as pentose phosphate, $NAD^+$ salvage, and transsulfuration; (v) depletion of glucose, 3-HB, and glycerol in serum from old mice is consistent with reduced supply of these fuel substrates from liver and adipose tissue to other organs; (vi) aging in mice favors up-modulation of glucose metabolism in cardiac and skeletal muscle as well as in liver, where gluconeogenesis and urea cycle are also enhanced, as well as similar but less pronounced pattern in brain. We then evaluated the associations between metabolites and phenotypic covariates (energetic cost of walking/running, RER, and frailty) by selecting the 24 most representative metabolites independently from age and organ. The main results showed that (vii) variance in energetic cost and RER can be explained by a distinct signature pattern of metabolic remodeling in the liver (e.g. mixed glucose, lipid, and amino acid metabolism), cardiac and skeletal muscle (e.g. glucose and lipids), and brain (mixture of amino acids in addition to glucose and ketone body catabolism). The enhancement in energy demand elicited an increasingly mixed substrate utilization (glucose, lipids, amino acids) leading to a rise in

oxidative stress in organs that have lower antioxidant capacity relative to the heart and liver. (viii) Frailty, as a complex phenotype less directly related to metabolism, did not show a consistent metabolic pattern compared to the other two physiological metrics.

Overall, the ensemble of data obtained from the cross-sectional correlation linking key metabolites to relevant phenotypes identified in this study is consistent with signatures of biological aging. These signatures correspond to patterns of metabolic remodeling that characterize age-associated changes in functional trajectory of mice coping with the increasing energetic cost to maintain health, energy, and redox balance. While many of these molecular alterations are likely to promote aging and aging-associated functional decline, other alterations are expected to maintain functional capacity with age.

The search for the biological mechanisms underlying healthy aging has shifted the focus of this research field toward the extension of healthspan and lifespan, and to delaying the onset of disease (*Longo and Panda, 2016*; *de Cabo and Mattson, 2019*; *Mitchell et al., 2019*; *Di Francesco et al., 2018*; *Aon et al., 2020*). This shift is helping to understand the differences between *biological* and *chronological* aging, underscoring that aging is not just the passage of time while emphasizing the notion that individuals age at a different pace (*Ferrucci et al., 2020*). For instance, the rate of change and onset of increase or decline in functional, behavioral, morphometric, and biochemical organismic indicators exhibit distinct time course as a function of age. This behavior agrees with the notion of *biological* rather than *chronological* aging, thereby showing that aging follows the trajectory of the organism and not merely the passage of time.

Body composition plays an important role in determining metabolic health, as can be judged by fat and lean content. Consistent with the body weight trajectory, younger male mice have a higher percent fat content than old animals due to larger visceral fat depots. Interestingly, a positive association was observed between fasting glucose levels and percent body fat, which, in turn, correlates with extended lifespan in mice (*Mitchell et al., 2016*). Age-associated regional distribution among fat depots has strong metabolic implications. Visceral fat accumulation is associated with insulin resistance (*Girard and Lafontan, 2008*; *Hardy et al., 2012*), while subcutaneous fat plays a role in reducing insulin levels and improving insulin sensitivity (*Tran et al., 2008*; *Shuster et al., 2012*). Here, the increase in body fat percentage from young to adult mice was coincident with a greater subcutaneous/visceral fat ratio, which, in turn, was associated with a significant drop in circulating insulin levels and HOMA-IR. Noteworthily, the oldest mice had the highest ratio of subcutaneous-to-visceral fat and a likelihood of being insulin sensitive (lowest HOMA-IR). We surmise that in order to preserve body fat for insulation, the oldest mice fed ad libitum must eat a significantly greater amount of food despite their low body weight, resulting in negative energy balance. Since the food intake in proportion to body weight is increased and the energy expenditure is decreased in these animals, they are able to rely less heavily on fat oxidation and more on carbohydrate utilization for energy needs, yielding RER values close to 1.0 in the dark phase.

In vivo imaging using MRI and microCT enabled a precise monitoring of body composition and bone density. These assessments illustrated the marked and significant decline in bone thickness and mineral density, and the reduction in fat and relative increase in lean body composition as a function of age (*Figure 2A–C*). These measurements are also of great interest in human aging; for example, the relationship between brain structure and composition with gait in humans has been described (*Rosso et al., 2017*; *for the Health ABC Study et al., 2016*). In vivo imaging techniques were recently reviewed in the context of their importance to studying aging (*Dall'Ara et al., 2016*).

Previously, it was demonstrated that the frailty index, as measured with the Mouse Clinical FI, increases with age and is significantly reduced in response to CR in male C57BL/6J mice (*Whitehead et al., 2014*; *Kane et al., 2016*). Frail animals lack resilience, causing them to die earlier due to the rapid deterioration of several tissue/organ systems. In this study, frailty was associated with an age-dependent decline in physical strength, tail height, gait speed, and circulating glucose levels. The rising energetic cost of walking is associated with steeper decline in gait speed in the elderly (*Schrack et al., 2016*). The results reported herein support these findings. Specifically, the $VO_2$ / gait speed ratio (in mL $O_2$/Kg/m) is energetically and physiologically consistent with the energetic cost of, for example, walking or running (*Figure 2D*), to the extent that the cost of running depends upon the animal's size rather than the speed at which an animal runs (*Taylor et al., 1970*; reviewed in *Schmidt-Nielsen, 1984*). Although the impact of age on skeletal muscle mass or fiber type was not assessed or considered in the present study, tendon length and other aspects of

muscle/tendon elasticity have been reported to be a major contributor of the biomechanics of work economy (*Higham et al., 2011*; *Wilson and Lichtwark, 2011*), and plans are underway to examine whether such changes in muscle/tendon properties are evident in old animals.

Biological aging represents a complex manifold and, in terms of the organism, widespread phenomenon. This is revealed by the intersection between its macroscopic manifestations such as physical performance and functional decline, with biological mechanisms deeply rooted in cellular energetics, repair, turnover, and detoxification. Our work highlights some of these potential functional/molecular interactions and demonstrates that these mechanisms operate differently at the level of an organ and its function, leading to a distinct and selective age-mediated impact, in turn, consistent with the differential metabolic remodeling observed.

### Limitations of the study

This was a cross-sectional study in male C57Bl/6N mice rather than a longitudinal study of male and female mice of different genetic background which would have accounted for genetic variability and sex. Resilience as well as immune and blood pressure measurements (*Bellantuono et al., 2020*) were not carried out in this study. Future work will require, at least in part, untargeted proteomics and assessment of epigenetic marks in various organs. The age- and frailty-dependent reduction in gait speed and associated energetic cost indicates the need to assess mitochondrial function using high-resolution respirometry in permeabilized tissues and isolated mitochondria in parallel with skeletal muscle ultrastructure by electron microscopy. Other approaches include incorporating MRI of the brain to identify changes in brain structure and volume over time that correlate with collected variables such as motor function, as well as assessments of cognition.

Further integration of omics in an all-inclusive phenotypic investigation would be beneficial in order to unveil new mechanistic insights potentially translatable to humans. Examples of multi-tissue omics analysis include a recent transcriptome analysis in rats that revealed a top list of genes potentially important in aging (*Shavlakadze et al., 2019*).

## Materials and methods

**Key resources table**

| Reagent type (species) or resource | Designation | Source or reference | Identifiers | Additional information |
|---|---|---|---|---|
| Strain, strain background (*M. musculus*, males) | C57BL/6NCrl | Charles River Laboratories | Strain Code 701 | a.k.a. NIA Aging Mouse Colony |
| Biological sample (*M. musculus*) | Brain from C57BL/6NCrl | B6MAL | | Metabolomics analysis, see lines 249–352 |
| Biological sample (*M. musculus*) | Heart from C57BL/6NCrl | B6MAL | | Metabolomics analysis, see lines 249–352 |
| Biological sample (*M. musculus*) | Liver from C57BL/6NCrl | B6MAL | | Metabolomics analysis, see lines 249–352 |
| Biological sample (*M. musculus*) | Serum from C57BL/6NCrl | B6MAL | | Metabolomics analysis, see lines 249–352 |
| Biological sample (*M. musculus*) | Skeletal muscle from C57BL/6NCrl | B6MAL | | Metabolomics analysis, see lines 249–352 |
| Commercial assay or kit | Mouse insulin ELISA kit | Crystal Chem, Inc | Cat# 90080 | |
| Software, algorithm | MetaboAnalyst (versions 3.0, 4.0) | Web-based resource (McGill University, Canada) | https://www.metaboanalyst.ca/MetaboAnalyst/faces/home.xhtml | |
| Software, algorithm | OriginLab 2018 | OriginLab corporation | https://www.originlab.com | |
| Software, algorithm | Prism 7.0 | GraphPad | RRID:SCR_015807 | |
| Software, algorithm | Canvas Draw six for macOS | Canvas GFX | RRID:SCR_014288 | |
| Software, algorithm | Microsoft Excel (version 16.19) | Microsoft Corp. | https://www.microsoft.com/en-gb/; RRID:SCR_016137 | |

*Continued on next page*

*Continued*

| Reagent type (species) or resource | Designation | Source or reference | Identifiers | Additional information |
|---|---|---|---|---|
| Other | TSE MotoRater system | TSE Systems | | |
| Other | 7T preclinical MRI scanner | Bruker Biospec 70/30 | | |
| Other | Oxymax Open Circuit Indirect Calorimeters | Columbus Instruments | http://www.colinst.com/docs/OxymaxBrochure.pdf | |
| Other | Modular Metabolic Treadmill | Columbus Instruments | | |
| Other | Minispec Whole Body Composition Analyzer LF90 | Bruker Optics | https://www.bruker.com/products/mr/td-nmr/minispec-lf-series.html | |
| Other | Skyscan 1076 micro-CT scanner | Bruker | | |

## Husbandry, diets, and dietary intervention in mice

Male C57BL/6N mice were obtained from the NIA Aging Mouse Colony (Charles River Laboratories, Kingston, NY). Animal rooms were maintained at 22.2 ± 1°C and 30–70% humidity. The lights were turned off at 6:00 PM and back on at 6:00 AM each day. Mice were group-housed up to four per cage with ad libitum access to food (Teklad Global 18% Protein Extruded Rodent Diet, #2018S, Envigo, Frederick, MD) and drinking water. In the case where an animal's cage mate died during the course of the study, animals were not rehoused and, instead, were provided a scientific exception to justify the single housing of a social species. All animals were provided shepherd shacks and nestlets for enrichment. All animal protocols were approved by the Animal Care and Use Committee of the National Institute on Aging, NIH. All tests in this cross-sectional cohort of animals were performed by non-blinded investigators (MP, SJM, MK, KWF). There were three age groups of mice, in months: Young, 3–8 (n = 23); Adult, 13–23 (n = 27); and Old, 27–36 (n = 23). Two mice, one adult (A) and one old (O), died during experimentation, bringing the final number of groups A and O mice to 26 and 22, respectively.

## Body composition

Unanesthetized mice were placed in the body comp clear plastic tube of a Bruker's Minispec Whole Body Composition Analyzer LF90 (Bruker Optics, Billerica, MA, USA). This low-field nuclear magnetic resonance device acquires and analyzes Time Domain-NMR signals from all protons in the entire sample volume and measures body fat, free body fluid, and lean tissue content. Young (n = 8), adult (n = 22), and old (n = 18) mice were studied.

## Wire hang test

Each mouse was tested for forelimb strength using the wire hanging test. Prior to testing, mice were trained and acclimatized by placing them on the wire of the hanging bar apparatus for several min. The test can be used to measure decline in strength and neuromuscular function with age and consisted of measuring the time before the mouse falls from the hanging wire. The test was repeated two more times with approximately 30 min break between trials for each mouse.

## Mouse gait speed and tail height analysis

Mice were acclimated to the TSE MotoRater system (TSE Systems, Inc Chesterfield, MO) for a week before gait and tail height recording as described (*Bair et al., 2019*). In brief, during gait assessment, the home cage was placed at the opposite end of the corridor to motivate the mouse to walk. A valid trial consisted of having the mouse walking the corridor length without stopping or turning in the opposite direction in the 70 cm mid-runway filming area. A custom in-house Python script was used to calculate mid- and maximum tail height, which was tracked by a semi-automated computer vision software package (TSE Systems). The Python script will be provided upon request for MotoRater system user. Gait speed was analyzed by PROC GLM model with SAS 9.4.

## Frailty index

Mice are assessed for frailty using the Howlett and Rockwood clinical frailty index, which includes 31 different parameters (*Feridooni et al., 2015*; *Bellantuono et al., 2020*). In brief, the frailty index was calculated by summing the number of deficits in factors related to integument, physical/muscu-loskeletal, vestibulocochlear/auditory, ocular/nasal, digestive/urogenital, respiratory, and aspects of physical discomfort for each mouse and dividing by the total number of possible deficits. Along with these indices, mouse weight and body temperature were also scored. A higher frailty index, with a potential score ranging from 0 to 1, is associated with lower survivability in mice (*Kane et al., 2016*).

## Magnetic resonance imaging

A 7T preclinical magnetic resonance imaging scanner (Bruker Biospec 70/30, Ettlingen, Germany) was used to measure volumes of visceral and subcutaneous fat from respiratory-gated axial fast spin echo images spanning the entire cranial-caudal extent of both kidneys. The ratio of total subcutane-ous to visceral fat volume was calculated and compared as a function of age.

## Bone imaging

Mice were anesthetized using avertin (tribromoethanol; Sigma-Aldrich, St-Louis, MO) and tibia were imaged by three-dimensional micro–computed tomography with a Bruker Skyscan 1076 micro-CT instrument for measurement and analysis of cortical and trabecular bones as previously described (*Mercken et al., 2014*). In brief, tibias were scanned at high resolution (1208 × 1080 pixels) with an isotropic voxel size of 18 µm and at 48 kV, 200 mA (0.5 mm Al filter). Microstructural properties of tibial cortical and trabecular bone were assessed at the distal metaphysis for trabecular parameters and the mid-diaphysis for cortical parameters, following the nomenclature guidelines outlined by *Dempster et al., 2013*. Trabecular bone was delineated via manual tracing and interpolation of tra-beculae in a region of interest 0.25 mm to 1.75 mm proximal to the distal tibial growth plate. For cortical bone parameters, analysis was performed at a volume of interest beginning at the mid-diaphysis of the tibia and extending 0.6 mm distally. Percent fat and percent lean tissue around the tibia was also quantified by micro-CT and normalized per unit of body weight.

## Metabolic treadmill

Mice were acclimated on the metabolic treadmills (Columbus Instruments International, Columbus, OH) at 5 m/min for 30 min the day prior to testing to ensure familiarity. On the day of testing, mice ran at their age group's natural walking gait speed, assessed by the TSE MotoRater in earlier experi-ments, for 45 min. External motivators such as electric shock from an electrified metal grid located near the moving belt and/or pocking are used to entice mice to run. Mice unwilling or incapable to continue running despite receiving electric shocks more than five consecutive times within few sec-onds meet the criterion for exhaustion and the testing ends.

## In vivo metabolic assessment

The metabolic rate of each mouse was assessed by indirect calorimetry in open-circuit Oxymax chambers using the Comprehensive Lab Animal Monitoring System (CLAMS; Columbus Instruments, Columbus, OH, USA). Mice were single housed with ad libitum access to water and food, and main-tained at 20–22°C under a 12:12 hr light:dark cycle (light period 0600–1800). Food and water level in each chamber was checked once on a daily basis at approximately 8:30 AM and the amount of food left in the cage from the previous 24 hr of chamber time was collected and measured. All mice were acclimatized to monitoring cages for 12 hr prior to recording. Sample air was dried and passed through an oxygen sensor for determination of oxygen content. Oxygen consumption was deter-mined by measuring oxygen concentration in air entering the chamber compared with air leaving the chamber. The sensor was calibrated against a standard gas mix containing defined quantities of oxygen, carbon dioxide and nitrogen. Constant airflow (0.5 L/min) was drawn through the chamber and monitored by a mass-sensitive flow meter. The concentrations of oxygen and carbon dioxide were monitored at the inlet and outlet of the sealed chambers to calculate oxygen consumption. Measurement in each chamber was recorded for 30 s at 30 min intervals for a total of 72 hr. Ambula-tory activity (both horizontal and vertical) was also monitored by dual axis detection using infrared

photocell technology. Metabolic parameters were analyzed using CalR software as previously described (*Mina et al., 2018*). Young (n = 8), adult (n = 22), and old (n = 18) mice were studied.

### Blood and serum markers and HOMA-IR calculation

Glucose concentrations in blood were measured from the submandibular vein in 6 hr fasted mice with the Blood Glucose Monitoring System Breeze 2 (Bayer, Mishawaka, IN). Coagulated blood was centrifuged at 12,000 x *g* at 4˚C for 10 min. Serum was aliquoted and kept frozen at −80˚C. Insulin levels were determined according to the manufacturer's protocol (Crystal Chem, Inc, Downers Grove, IL). Homeostasis model assessment-insulin resistance (HOMA-IR) was calculated to assess group differences in insulin resistance using the HOMA2 Calculator software available from the Oxford Centre for Diabetes, Endocrinology and Metabolism Diabetes Trials Unit website (http://www.dtu.ox.ac.uk/homacalculator/index.php).

### Metabolomics analysis

Metabolomic analysis was performed by the West Coast Metabolomics Center at UC Davis (Davis, CA) in brain, heart, liver, skeletal muscle, and serum from nonfasted animals as previously described (*Mitchell et al., 2018*; *Mitchell et al., 2016*). In brief, tissue and serum were extracted in an acetonitrile:isopropanol:water (3:3:2) solution, vortexed, centrifuged, and the supernatants aliquoted for downstream analysis. After a series of evaporation and reconstitution steps in 50% acetonitrile, internal standards (C08-C30, fatty acid methyl esters) were added to the dried sample, which was then derivatized for trimethylsilylation of acidic protons. Data were acquired using the method as described by *Fiehn, 2008* and summarized by *Mitchell et al., 2016*. In brief, metabolites were measured using a rtx5Sil-MS column (made of 95% dimethyl, 5% diphenyl-polysiloxane coated on fused silica; Restek Corporation; Bellefonte PA) protected by an empty guard column. This chromatography method yields excellent retention and separation of primary metabolite classes (amino acids, hydroxyl acids, carbohydrates, sugar acids, sterols, aromatics, nucleosides, amines, and miscellaneous compounds) with arrow peak widths of 2–3 s and very good within-series retention time reproducibility of better than 0.2 s absolute deviation of retention times. The mobile phase consisted of helium, with a flow rate of 1 mL/min, and injection volume of 0.5 μL. The following mass spectrometry parameters were used: a Leco Pegasus IV mass spectrometer with unit mass resolution at 17 spectra s-1 from 80 to 500 Da at −70 eV for elution of metabolites. As a quality control, for each sequence of sample extractions, one blank negative control was performed by applying the total procedure (e.g. all materials and plastic ware) without biological sample. Result files were transformed by calculating the sum intensities of all structurally identified compounds for each sample, and subsequently dividing all data associated with a sample by the corresponding metabolite sum. The resulting data were multiplied by a constant factor in order to obtain values without decimal places. Intensities of identified metabolites with more than one peak (e.g. for the syn- and anti-forms of methoximated reducing sugars) were summed to only one value in the transformed data set. The original non-transformed data set was retained. Relative metabolite levels represent the MS peak amplitude normalized with respect to the total metabolites returned, but disregarding unknowns that might potentially comprise artifact peaks or chemical contaminants.

### Quantification and statistical analysis

No statistical method was used to predetermine sample size. GraphPad Prism version 7.0 was used to determine whether data sets passed the D'Agostino and Pearson normality test ($\alpha = 0.05$). Those that failed included body composition data (NMR-generated morphometric analysis), physical performance/motor coordination results and MRI quantitative data, whereas metabolic outputs, locomotor activity between light/dark cycles, and bone imaging followed a normal data distribution. One-way ANOVA or Kruskal-Wallis H test was performed to determine if there were statistically significant differences between the three age groups for any given outcome. All regression models were either quadratic or linear regressions based on data's fit. The $R^2$, F-statistic and p-value are calculated through these models. Outliers were not omitted. As the mice aged, the different assessments took place. At times, the experiments were 3 or 6 months apart from previous assessments, which is why 36-month-old mice were enrolled in some measured outcomes, and not in others.

The polynomial variables for all curvilinear lines fitted to parameters are as follow:

*Figure 1K* (Glucose vs Age): $-0.11821x^2 + 3.22945x + 149.09292$
*Figure 2A* (Body Weight vs Age): $y = -0.046141x^2 + 1.632253x + 25.087950$
*Figure 2A* (%Fat vs Age): $y = -0.040847x^2 + 1.266635x + 11.621481$
*Figure 2A* (%Lean vs Age): $y = 0.021048x^2 + -0.593099x + 62.762510$
*Figure 2C* (%Lean Tibia vs Age): $y = 0.003239x^2 + -0.127683x + 3.903823$
*Figure 2D* (Energetic cost vs Age): $0.8025x^2 + -18.182x + 572.9071$
*Figure 2D* (Heat vs Age): $-0.001791x^2 + 0.011949x + 5.174154$
*Figure 2K* (RER vs Age, dark phase): $y = 3.390e\text{-}04x^2 + -1.088e\text{-}02x + 9.216e\text{-}01$

Multiple linear regression analysis was used to examine the determinants of frailty index and gait speed, including, as covariates:

1. Frailty index vs. age + gait velocity + BW + mean tail height + latency to fall from wire hang + glucose + HOMA-IR;
2. #1 + %Lean mass + subcutaneous fat/visceral fat ratio;
3. #1 + trabecular BMD + cortical thickness + %fat around tibia/BW + %lean around tibia/BW;
4. #1 + Metabolic treadmill outputs (energetic cost + mean heat + RER).

The models were fitted with all explanatory variables as well as models that included interactions of Age with each of the other explanatory variables. All the interactions were then tested to assess whether they added anything to the original model (with only main effects). In almost all cases, the interactions did not improve the models. A backward elimination was then performed to obtain the final regression model. There was likely not enough power to detect significant interactions; however, these findings do not mean that significant interactions do not exist, but simply that we were not able to detect any significant interactions due largely to the limited size of this dataset. All analyses were performed with R Version 4.0.2 (*R Development Core Team, 2020*) with RStudio Version 1.3.1073.

Although dozens of different health parameters were evaluated on the same animals, these were largely the results of independent experiments to test distinct hypotheses. Therefore, no correction for multiple comparisons to the p-values was made as each experiment was considered to be a separate question. p Value $\leq 0.05$ was considered statistically significant.

The software MetaboAnalyst, an integrated web-based platform for comprehensive analysis of metabolomics data, versions 3.0 (*Xia and Wishart, 2016*) and 4.0 (*Chong et al., 2019*), was utilized. Univariate (ANOVA), clustering (heat map, correlation matrix) and multivariate (partial least square discriminant; PLSD) statistical analyses were applied to the metabolite profiles (See *Figures 3–6*, *Figure 3—figure supplement 1*, *Figure 4—figure supplement 1*, *Figure 6—figure supplement 1*). PLSD analysis revealed that all tissues exhibited good separation as a function of age, and a subset of 49 metabolites responsible for the separation was identified. The autoscaling function of MetaboAnalyst used to normalize the metabolomics data closely resembles the Z ratio expression and requires the detection and removal of outliers.

One-way ANOVA (Prism 7.0, GraphPad Software, San Diego, CA, USA) was performed with Tukey's multi-comparison test for the three age groups (young, adult, old) within each tissue. The statistical significance of the effect of age was evaluated for those metabolites that exhibited a significant difference in at least one of the tissues analyzed. Venn diagrams were plotted with Canvas-Draw v. 5.0 for Mac (Canvas GFX, Inc, Plantation, FL).

## Acknowledgements

This study was supported by the Intramural Research Program of the National Institute on Aging, National Institutes of Health, Baltimore, MD, USA. We are grateful to the Comparative Medicine Section of the NIA, NIH for their exceptional animal care. We thank the NIA Aging Rodent Colony for providing aging mice for this study as part of their aging mouse resource.

# Additional information

## Funding

| Funder | Grant reference number | Author |
|---|---|---|
| National Institute on Aging | AG000335-04 | Rafael de Cabo |

The funders had no role in study design, data collection and interpretation, or the decision to submit the work for publication.

## Author contributions

Michael A Petr, Data curation, Formal analysis, Investigation, Writing - original draft, Writing - review and editing; Irene Alfaras, Formal analysis, Investigation, Project administration, Writing - review and editing; Melissa Krawcyzk, Woei-Nan Bair, Data curation, Formal analysis, Investigation; Sarah J Mitchell, Data curation, Investigation, Writing - review and editing; Christopher H Morrell, Data curation, Formal analysis, Writing - review and editing; Stephanie A Studenski, Edward G Lakatta, Conceptualization, Writing - review and editing; Nathan L Price, Kenneth W Fishbein, Data curation, Formal analysis, Investigation, Writing - review and editing; Richard G Spencer, Formal analysis, Writing - review and editing; Morten Scheibye-Knudsen, Conceptualization, Formal analysis, Writing - review and editing; Luigi Ferrucci, Conceptualization, Resources, Writing - review and editing; Miguel A Aon, Michel Bernier, Conceptualization, Data curation, Formal analysis, Writing - original draft, Writing - review and editing; Rafael de Cabo, Conceptualization, Resources, Data curation, Formal analysis, Supervision, Funding acquisition, Investigation, Methodology, Project administration, Writing - review and editing

## Author ORCIDs

Luigi Ferrucci http://orcid.org/0000-0002-6273-1613
Michel Bernier https://orcid.org/0000-0002-5948-368X
Rafael de Cabo https://orcid.org/0000-0003-2830-5693

## Ethics

Animal experimentation: This study was performed in strict accordance with the recommendations in the Guide for the Care and Use of Laboratory Animals of the National Institutes of Health. All of the animals were handled according to approved institutional animal care and use committee (ACUC) protocol of the National Institute on Aging (Protocol Number: TGB-277-2022).

## Decision letter and Author response

Decision letter https://doi.org/10.7554/eLife.62952.sa1
Author response https://doi.org/10.7554/eLife.62952.sa2

# Additional files

## Supplementary files

• Transparent reporting form

## Data availability

This study did not generate datasets or code.

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
