## [Decision Letter]

**Acceptance summary:**

This is a thorough characterization of phenotypes and metabolomic profiles across male C57Bl/6N mice ranging from 3-36 months coupled with a meta-analysis transcriptomic and metabolomics data from mice fed ad libitum or maintained chronically on calorie restriction (CR). The new analyses linking organ-specific patterns of increased abundance or depletion of metabolites with primary functional measures (energetic cost of walking, RER, and FI) are phenomenal. The interpretations of differences in small metabolite signatures in energetic cost and RER vs FI are insightful. Overall, the authors have turned a compelling submission to an exceptional manuscript that is likely to advance the field.

**Decision letter after peer review:**

Thank you for submitting your article "A cross-sectional study of functional and metabolic changes during aging through the lifespan in male mice" for consideration by *eLife*. Your article has been reviewed by 3 peer reviewers, one of whom is a member of our Board of Reviewing Editors, and the evaluation has been overseen by Jessica Tyler as the Senior Editor. The reviewers have opted to remain anonymous.

The reviewers have discussed the reviews with one another and the Reviewing Editor has drafted this decision to help you prepare a revised submission.

Summary:

The authors performed a cross-sectional analysis of (1) physiologic and functional tests, and (2) untargeted metabolomics in multiple organs and tissues in male C57Bl/6N mice ranging from 3-36 months of age; and (3) transcriptomic and untargeted metabolomics data from liver of 24-mo-old mice fed ad libitum or maintained chronically on calorie restriction (CR). The impact of the present study is derived from the expansive breadth and depth of assessments performed.

Essential revisions:

A major limitation is use of a single sex and inbred strain, and lack of longitudinal assessment would strengthen study. While new experiments to address these deficiencies could enhance the study, all reviewers agree that these are addressed in discussion, are not essential for publication at this stage, and would require a timeline likely beyond the scope here.

Another over-arching concerned raised by reviewers is that the study is appears to be two or three separate studies combined together in one manuscript (healthspan assessment; metabolomics; caloric restriction). On one hand, this strengthens the study by providing an substantial amount of data. On the other hand, this also highlights a weakness: the study lacks cohesion, is highly descriptive, and does not make a convincing argument for causation (or even association with health). The major revision that is needed is a more cohesive narrative that unifies these areas.

1. A major element missing is omission of analyses and discussion linking healthspan assessment (physical function, frailty index, energetics, work economy) to the omics. If metabolomics and function performed in the same mice, it would be valuable to identify small metabolite signatures associated with gait speed or frailty index, etc.

2. Rationale for including CR data must be improved as cohorts, design, and measures are not consistent (different functional assessments, RNAseq included, etc).

3. Previous cross-sectional health studies in mice should be referenced for context (e.g. DOI: 10.1007/s11357-013-9589-9, 10.18632/aging.101059).

4. Figure 1. frailty and age are collinear; if age is used as a covariate is the effect of frailty removed?

5. Figure 2.

a. Legend and figure not aligned.

b. Panels 2E,F,G require clarification as interpretation is not intuitive. It looks as if higher gait velocity is associated with higher frailty index in old animals. Please consider alternative presentation format and check alignment with legend and text.

c. If curvilinear line fitted to parameters, please reference polynomial variables in methods.

d. Panel D – please report if differences in heat production and work economy are maintained after adjusting for lean mass. Also note if values generated differ over the entire activity episode or if energetic cost was initially similar across ages and then waned in old animals over time (if the latter, could there be a role for fatigue etc)

e. Consider segregating the Figure 2J and K data by light and dark cycle in addition to 24hr avg to better visualize the diurnal changes in these variables.

6. Interpretation of data in oldest group.

a. Additional comment on the extent that changes in the oldest group are an age effect per se versus negative energy balance effect are warranted (i.e. animals are in a gradual, chronic decline characterized by loss of weight and adiposity) – examples, reduction in HOMA-IR and circulating fatty acids. This deserve some discussion and delineation as such findings appear at odds with what is often thought to define metabolic changes with aging in mammalians, including humans, but appears periodically in studies using older rodents.

b. Authors propose "survivorship bias" in glucose homeostasis, but this is not consistent with other physiologic systems.

7. Describe feeding, collection of 24hr food intake. Comment of role of feeding in relation to energetics and raised RER in dark phase for oldest mice.

[Editors' note: further revisions were suggested prior to acceptance, as described below.]

Thank you for resubmitting your work entitled "A cross-sectional study of functional and metabolic changes during aging through the lifespan in male mice" for further consideration by *eLife*. Your revised article has been evaluated by Jessica Tyler (Senior Editor) and a Reviewing Editor.

This is a thorough characterization of phenotypes and metabolomic profiles across male C57Bl/6N mice ranging from 3-36 months coupled with a meta-analysis transcriptomic and metabolomics data from mice fed ad libitum or maintained chronically on calorie restriction (CR).

In general, this is a meritorious and comprehensive examination of phenotypic, functional, and metabolomic profiles across the lifespan in male C57BL/6J mice of anticipated interest to those in biology of aging, geroscience, and preclinical interventions testing. Though primary study design is cross-sectional and in a single mouse strain and sex, the impact of the present study is derived from (1) the expansive breadth and depth of assessments performed, (2) the pairing of functional and metabolomic profiles in a single study, and (3) the exploration of CR compared with ad libitum fed animals.

The manuscript has been improved but there are some remaining issues that need to be addressed, as outlined below:

1. Reviewers requested authors evaluate the associations between key metabolites and frailty or functional measures. Please address.

2. The new Discussion section on metabolome and phenotypes is lengthy and somewhat reaching; please shorten.

3. Though rationale for including the CR data in introduction is improved, there is remaining reviewer concern that transcriptomic data in the CR study vs main study is imbalanced. Do the metabolomes of the CR animals look more like those of the adult animals in the main study?

Reviewer #1:

The authors were responsive to reviewer suggestion; most revisions and select omissions are justified. The revised Figure 2, additional analyses (e.g. age added to frailty model; diurnal patterns in RER ), text to add rational for CR meta-analysis, and clarifying text have improved readability and interpretation of findings. Some concerns either were not or could not be fully addressed. Overall the revisions have improved the quality and merit of the manuscript. This is a nice paper that is anticipated to have high impact.

1. Reviewers requested authors evaluate the associations between 'omics' and phenotypes (e.g. frailty); this was not addressed directly. Do key metabolites (e.g. pentose phosphate, NAD+ salvage and transsulfuration; depletion of glucose, 3-HB, and glycerol in serum) explain the variability in frailty with and without adjustment for chronologic age?

2. In lieu of modeling, the authors provided a lengthy and somewhat reaching review of multi-omics and phenotypes of aging in the discussion. A detailed description may be of interest in a dedicated review, but the Discussion section would benefit from a concise presentation.

Reviewer #2:

I found overall that the manuscript was much improved, and the majority of the comments were well addressed.

However, I still feel the rationale for including the CR data is not strong, and I would recommend it be removed. As there was no transcriptomic data for the main study, the CR transcriptome feels especially out of place. It would have also been helpful for the authors to show how the metabolomes changes in their cross sectional and CR studies. Do the metabolomes of the CR animals look more like those of the adult animals in the main study?

In addition, while the authors expanded on their metabolomics interpretations some, they did not correlated their metabolite values to the functional values measured.

---

## [Author Response]

Essential revisions:A major limitation is use of a single sex and inbred strain, and lack of longitudinal assessment would strengthen study. While new experiments to address these deficiencies could enhance the study, all reviewers agree that these are addressed in discussion, are not essential for publication at this stage, and would require a timeline likely beyond the scope here.

We are thankful to the reviewers for their appreciation of the study despite its inherent limitations as highlighted in the Discussion. We and others have demonstrated that genetic heterogeneity and sex can have dramatic impact on functional and phenotypic aging and the effectiveness of anti-aging interventions in mice (Mitchell et al., 2016; Gonzalez-Freire et al., 2020). The knowledge gained from the current cross-sectional study led us to the design and early implementation of the Study of Longitudinal Aging in Mice (SLAM) in which inbred C57BL/6J and outbred UM-HET3 mice of both sexes are longitudinally evaluated for functional, phenotypic and biological health, and collection of biospecimens is conducted throughout their lifespan (Palliyaguru et al., 2020). The SLAM study is at an early stage but eventually will provide the tools needed to identify and characterize phenotypic and biological predictors of mouse aging and age-associated conditions, assess whether these changes are consistent with alterations observed in human aging, and evaluate the utility of mouse models for anti-aging interventions in the clinic.

Another over-arching concerned raised by reviewers is that the study is appears to be two or three separate studies combined together in one manuscript (healthspan assessment; metabolomics; caloric restriction). On one hand, this strengthens the study by providing an substantial amount of data. On the other hand, this also highlights a weakness: the study lacks cohesion, is highly descriptive, and does not make a convincing argument for causation (or even association with health). The major revision that is needed is a more cohesive narrative that unifies these areas.

We thank the editors for this important guidance. We have made a significant effort to offer a more cohesive narrative of the various aspects of the study and provide arguments as to why multi-organ untargeted metabolomics and transcriptomics analyses along with a nutritional intervention were carried out in order to fill a gap of knowledge on the effect of age on key phenotypic aging domains associated with health and mortality. See our answers to Q1 and Q2.

1. A major element missing is omission of analyses and discussion linking healthspan assessment (physical function, frailty index, energetics, work economy) to the omics. If metabolomics and function performed in the same mice, it would be valuable to identify small metabolite signatures associated with gait speed or frailty index, etc.

The last few paragraphs of the Discussion section are now focused on presenting a narrative of what the ‘omics’ results mean with regard to the various healthspan outcomes while also describing the contribution our study makes to the field of aging.

2. Rationale for including CR data must be improved as cohorts, design, and measures are not consistent (different functional assessments, RNAseq included, etc).

While improved knowledge of the specific phenotypic and molecular changes that occur with aging is of considerable use to the research community, this type of characterization is further enhanced by the identification of specific alterations that are linked to lifespan extension through anti-aging interventions. One of the best non-pharmacological interventions that improves health and reduces mortality is calorie restriction (CR). CR contributes to negative energy balance by promoting loss of weight gain and reduction in adiposity, HOMA-IR, and circulating fatty acids compared to mice fed *ad libitum*. It is with that in mind that we performed a meta-analysis of an earlier study in which transcriptomic and untargeted metabolomics data were collected from liver of 24-mo-old C57BL/6 mice fed *ad libitum* or maintained chronically on CR (Mitchell et al., 2016). Although different functional assessments were carried out between the two studies, one common thread was our attempt to identify pathways enriched by both genes and metabolites that can be viewed as potential biomarkers of healthy aging in middle-aged mice. We found shared molecular and biochemical signatures between these two independent studies, which speaks to the robustness of these observations and highlights the importance of having the findings from the Mitchell et al. study to complement the current work. In the revised version of the manuscript, we have provided a more direct rationale for the inclusion of CR data and worked to better integrate the findings from this section into the discussion to more clearly demonstrate how these findings contribute to the overall conclusions of this work.

The rationale reads as follows: “In order to elucidate which of the specific phenotypic and molecular/metabolomic changes that occur with aging are altered in response to pro-longevity interventions, we conducted a multi-omics meta-analysis of the liver of 24-mo-old C57BL/6 male mice fed *ad libitum* (AL) or maintained chronically on 20% calorie restriction (CR) from one of our previous studies (Mitchell et al., 2016). These CR-fed mice had a ∼24% lifespan extension, which led us to hypothesize that pathways enriched by both genes and metabolites might be responsible for overcoming the impact of normal aging after lifelong CR and might be viewed as potential biomarkers of healthy aging in old mice. Indeed, many of the differences that are observed in animals on CR are the inverse of those observed in older cohorts of animals, while other age-related differences that are likely induced to preserve organ function with age are further enhanced in response to CR.”.

Moreover, we added the following statement: “Therefore, a key and consistent observation emerging from our two independent studies is the fact that the activation of redox-related pathways with age is observed in organs subjected to oxidative stress, such as liver and heart, which appears to be reinforced in CR-fed mice and may be a key driver of lifespan extension.”.

3. Previous cross-sectional health studies in mice should be referenced for context (e.g. DOI: 10.1007/s11357-013-9589-9, 10.18632/aging.101059).

The papers of Justice et al., 2014 and Fisher et al., 2016 have now been referenced.

4. Figure 1. frailty and age are collinear; if age is used as a covariate is the effect of frailty removed?

After adjusting for age, frailty is no longer associated with gait velocity (Figure 1G, p = 0.477) but it is still strongly associated with forelimb muscle strength (Figure 1H, p = 0.00088). After including age, frailty is only marginally associated with mean tail height (Figure 1I, p = 0.0792) while being still associated with maximal tail height (Figure 1J, p = 0.0495). The effects of frailty on muscle strength were maintained after correction for age, indicating the presence of independent risk factors beyond chronological age that could contribute to the reduction in physical performance in the oldest mice. A statement has been incorporated in the Results.

5. Figure 2.a. Legend and figure not aligned.

We are sorry for the confusion. The legends for Figure 2 E-G were mistakenly omitted from the original submission, which has been since corrected. See comments in R5b.

b. Panels 2E,F,G require clarification as interpretation is not intuitive. It looks as if higher gait velocity is associated with higher frailty index in old animals. Please consider alternative presentation format and check alignment with legend and text.

In the previous multiple regression analysis, we utilized a set of values of the covariates to obtain the plots, and variables that were not included in the plot were set to their means. However, this led to unrealistic interpretations such as having a frail old mouse with higher gait velocity than a young, nonfrail animal. The problem is that in certain regions the covariates can only take on a particular set of values (not the whole range of covariates). Therefore, to create more appropriate visualizations of the models, we set up a grid of intervals for the two variables being plotted.

Using the first model as an example:

FrailtyIndex ~ Age_months + GaitVelocity_ms + Glucose + HOMAIR + Age_months:GaitVelocity_ms

For the first plot shown in Figure 2E, Age had intervals: [0,5] [5,10] [10,15] [15,20] [20,25] [25,30] [30,35] and Gait Velocity had intervals: [0.05,0.15] [0.15,0.25] [0.25,0.35]. The intervals for the legend variable (Gait Velocity) were usually chosen so that there were about the same number of (or, at least, “enough”) observations in each group.

**Author response table 1. resptable1:** 

	Age group						
Gait group	[0,5]	[5,10]	[10,15]	[15,20]	[20,25]	[25,30]	[30,35]
[0.05, 0.15]	**0**	**0**	1	1	1	7	5
[0.15, 0.25]	3	4	4	13	6	4	6
[0.25, 0.35]	13	3	0	1	**0**	**0**	**0**

As can be seen in Author response table 1, certain combinations of these two group variables contain no observations. For example, we were unable to record old mice having high gait velocity (red font) nor were there any young mice walking at low gait velocity (blue font). Consequently, predictions should not be made in these regions. Other covariates in the model, Glucose + HOMA-IR, were set to their mean value in each grid cell to obtain model predictions. The x-values in the plot are the mean (Age) for each interval and the legend value is listed as the center of the intervals for that variable (e.g., 0.1, 0.2, and 0.3 for gait group). Although this is a cross-sectional study, the points were joined to help the reader see the association of Frailty with Age within the same gait velocity We are not suggesting that there is a causal association, just that animals with higher Age tend to have higher Frailty.

Although it does seem that there is a positive association between frailty index and gait speed in old mice (Figure 2E), the fact remains that many factors contribute to the frailty index score, as summarized in our recent review (Palliyaguru et al., 2019). Whether specific parameters related to ocular/nasal, digestive/urogenital and/or integumentary domains impact gait speed need to be firmly established.

The revised graphs depicted in Figure 2F,G are also only plotted over a reasonable region of the variables, with values of the other covariates in the model as mentioned above.

The legend for Figure 2 E-G has been incorporated into the document and it reads as follows:

“(E-G) Multiple regression analyses with frailty index (E, F) and gait velocity (G) as response variables. The modeling approach consisted of fitting a series of four regressions for each of the response variables. The explanatory variables included in the multiple regression modeling are depicted in Figure 2-Source data 1. (E) *left panel*, Effect of gait velocity on the relationship between frailty and age; *right panel*, Impact of age on the interaction between frailty and gait velocity. (F) *left panel*, Effect of bone cortical thickness on the relationship between frailty and age; *right panel*, Impact of age on the interaction between frailty and bone cortical thickness. (G) *left panel*, Effect of energetic costs on the relationship between gait velocity and age; *right panel*, Impact of age on the interaction between gait velocity and energetic costs. Although this is a cross-sectional study, the points were joined to help the reader see the various associations without suggesting that there is causal association; for example, that animals with higher age tend to have higher frailty within the same gait velocity (E, *left panel*).”

c. If curvilinear line fitted to parameters, please reference polynomial variables in methods.

As requested, we have provided the polynomial variables for all curvilinear lines fitted to parameters in the Methods section. These variables are as follow:

Figure 1K (Glucose vs Age): -0.11821x^2^ + 3.22945x + 149.09292

Figure 2A (Body Weight vs Age): y = -0.046141x^2^ + 1.632253x + 25.087950

Figure 2A (%Fat vs Age): y = -0.040847x^2^ + 1.266635x + 11.621481

Figure 2A (%Lean vs Age): y = 0.021048x^2^ + -0.593099x + 62.762510

Figure 2C (%Lean Tibia vs Age): y = 0.003239x^2^ + -0.127683x + 3.903823

Figure 2D (Energetic cost vs Age): 0.8025x^2^ + -18.182x + 572.9071

Figure 2D (Heat vs Age): -0.001791x^2^ + 0.011949x + 5.174154

Figure 2K (RER vs Age, dark phase): y = 3.390e-04x^2^+-1.088e-02x+9.216e-01

d. Panel D – please report if differences in heat production and work economy are maintained after adjusting for lean mass. Also note if values generated differ over the entire activity episode or if energetic cost was initially similar across ages and then waned in old animals over time (if the latter, could there be a role for fatigue etc)

The differences in work economy and heat production between the three age groups are maintained after adjusting for lean mass, although a much sharper drop in heat production between age groups can be observed after lean mass adjustment (compare bottom vs. upper right panels in Author response image 1). A statement has been incorporated in the Results section.

**Author response image 1. sa2fig1:** 

With regard to the second comment, the metabolic treadmill speed was set according to the MotoRater average gait speed for each age group (units in m/min. 8-mo, 14.4; 21-mo, 10.8; 31-37-mo, 7.1). In Author response image 2, we show that the running VO_2_ values did not differ over the entire activity episode (800 sec) across age, hence ruling out a role for fatigue in the higher energetic cost in the oldest mice. A statement has been incorporated in the Results section.

e. Consider segregating the Figure 2J and K data by light and dark cycle in addition to 24hr avg to better visualize the diurnal changes in these variables.

This is an excellent suggestion, indeed! The revised Figure 2J displays the diurnal changes in RER in the three age groups. The results indicate statistically significant differences in the dark cycle (D) between young (red) *vs.* old (black) and adult (blue) vs. old mice, with higher RER in the oldest group of mice (revised Figure 2K). These two new Figures are replacing the old ones as they provide stronger evidence of changes in RER between age groups. Note that no difference in RER could be observed in the light phase between the different groups of mice (r^2^ = 0.0763). To this end, a statement has been incorporated in the Results section.

6. Interpretation of data in oldest group.a. Additional comment on the extent that changes in the oldest group are an age effect per se versus negative energy balance effect are warranted (i.e. animals are in a gradual, chronic decline characterized by loss of weight and adiposity) – examples, reduction in HOMA-IR and circulating fatty acids. This deserve some discussion and delineation as such findings appear at odds with what is often thought to define metabolic changes with aging in mammalians, including humans, but appears periodically in studies using older rodents.

The reviewer has two main assumptions: (1) Aging is a ‘gradual’ decline that can be dissociated from negative energy balance; and (2) the energetic balance is negative with age as revealed by weight loss and adiposity.

Regarding point #1, our data do not support this assumption because of the fact that the increase in energetic cost is highly nonlinear (exponential) from ∼15 months of age, thus strongly associated with aging.

For point #2, one of the reasons for the negative energy balance in the oldest mice is the higher energetic cost of movement in general, as reflected by gait speed and other indicators. If the animals spend more energy for the same amount of movement/activity, this can explain, at least in part, the reduction in adiposity and weight. It follows that at some point during aging, the energy demand of movement/activity in general should offset the intake of food further contributing to the negative energy balance. Although the life extending properties of long-term CR is also associated with an increase in negative energy balance e.g., lower body weight gain combined with a reduction in adiposity, it also triggers a panoply of healthy physiological responses leading to enhanced health and survival. In this context, we surmise that the survival of the oldest mice (31-36 months of age) was accompanied by the slowing down of phenotypic aging domains (Gonzalez-Freire et al., 2020).

Within aging research, mice have been used to identify and characterize some of the key contributors of health and survival (e.g., fasting blood glucose, adiposity, body weight), and have served as a preclinical model to test anti-aging interventions (Gonzalez-Freire et al., 2020). In a soon-to-be published study, we compared age-associated metabolic indices across three increasingly genetically complex mouse strains (C57BL/6, HET3, and Diversity Outbred) with nonhuman primates and humans. The association of age with blood glucose and the relationship between blood glucose and mortality differs considerably between mice (in either strain) and what is observed among nonhuman primates and humans (Palliyaguru, Shiroma et al., submitted). These findings clearly illustrate the potential pitfall of generalizing age-associated changes in specific metabolic trajectories and biomarkers in mice *vs.* humans. More work is required to better evaluate the effectiveness of anti-aging interventions in the clinic.

b. Authors propose "survivorship bias" in glucose homeostasis, but this is not consistent with other physiologic systems.

The initial sentence has been amended to acknowledge the reviewer’s reservation. It now reads: “These results may reflect a survivorship bias of the long-living mice (Scheen, 2017), although other effects such as reduced body weight and age-associated regional distribution among fat depots could have profound metabolic implications vis-à-vis changes observed in fasting blood glucose and insulin resistance (Girard and Lafontan, 2008; Hardy et al. 2012; Tran et al., 2008; Shuster et al., 2012).”

7. Describe feeding, collection of 24hr food intake. Comment of role of feeding in relation to energetics and raised RER in dark phase for oldest mice.

We surmise that the reviewer refers to data collection while the mice were temporarily single housed in the metabolic testing chamber (Figure 2I-K). Food and water level in each chamber was checked once on a daily basis at approximately 8:30 AM and the amount of food left in the cage from the previous 24 h of chamber time was collected and measured. The Methods section has been amended to provide more details on this.

With regard to the role of feeding in relation to energetics and higher RER in the dark phase for the oldest mice fed *ad libitum*, we assume that in order to preserve body fat for insulation, old mice must eat a significant amount of food despite their low body weight, resulting in negative energy balance. Since the food intake in proportion to body weight is increased and the energy expenditure is decreased in these animals, they are able to rely less heavily on fat oxidation and more on carbohydrate utilization for energy needs, yielding RER values close to 1.0 in the dark phase. A statement has been added to the Discussion that highlights this important concept.

[Editors' note: further revisions were suggested prior to acceptance, as described below.]The manuscript has been improved but there are some remaining issues that need to be addressed, as outlined below:1. Reviewers requested authors evaluate the associations between key metabolites and frailty or functional measures. Please address.

Following the Reviewers’ suggestion, we devoted renewed efforts in assessing the strength of the links between metabolites and functional measures (e.g., energetic cost of walking, RER and frailty index) by investigating the organs and serum fine-tuning their metabolism and substrate selection to cope with the increased energetic demand of maintaining health and energy balance. A whole new section devoted to this topic is now included in Results and in the first paragraph of the Discussion. A new set of figures (Figure 6 and associated Figure 6-supplemental figure 1) has been generated as well to support these new findings.

2. The new Discussion section on metabolome and phenotypes is lengthy and somewhat reaching; please shorten.

In the latest iteration of the paper, we have removed sections in the discussion that did not contribute to the main narrative of the study and reworded the ‘metabolome-phenotype’ section by discussing the new findings regarding the associations between key metabolites and functional measures. The reference list has been updated accordingly.

3. Though rationale for including the CR data in introduction is improved, there is remaining reviewer concern that transcriptomic data in the CR study vs main study is imbalanced. Do the metabolomes of the CR animals look more like those of the adult animals in the main study?

We have decided to remove the CR data from the current manuscript and focus instead on the main thrust of the study that is aimed to carry out a cross-sectional multi-organs metabolomic analysis in male mice fed *ad libitum* and its pairing with functional outcomes. In doing so, no further analysis was carried out with the CR dataset.

Reviewer #1:The authors were responsive to reviewer suggestion; most revisions and select omissions are justified. The revised Figure 2, additional analyses (e.g. age added to frailty model; diurnal patterns in RER ), text to add rational for CR meta-analysis, and clarifying text have improved readability and interpretation of findings. Some concerns either were not or could not be fully addressed. Overall the revisions have improved the quality and merit of the manuscript. This is a nice paper that is anticipated to have high impact.1. Reviewers requested authors evaluate the associations between 'omics' and phenotypes (e.g. frailty); this was not addressed directly. Do key metabolites (e.g. pentose phosphate, NAD+ salvage and transsulfuration; depletion of glucose, 3-HB, and glycerol in serum) explain the variability in frailty with and without adjustment for chronologic age?

In this revised version of the manuscript, we devoted renewed efforts in assessing the strength of the links between metabolites and functional measures (e.g., energetic cost of walking, RER and frailty index) by investigating the organs and serum fine-tuning their metabolism and substrate selection to cope with the increased demand of maintaining health and energy balance. A whole new section devoted to this topic is now included in Results and in the first paragraph of the Discussion. A new set of figures (Figure 6 and associated Figure 6-supplemental figure 1) has been generated as well to support these new findings.

2. In lieu of modeling, the authors provided a lengthy and somewhat reaching review of multi-omics and phenotypes of aging in the discussion. A detailed description may be of interest in a dedicated review, but the Discussion section would benefit from a concise presentation.

In the latest iteration of the paper, we have removed sections in the discussion that did not contribute to the main narrative of the study and reworded the ‘metabolome-phenotype’ section by discussing the new findings regarding the associations between key metabolites and functional measures. The reference list has been updated accordingly.

Reviewer #2:I found overall that the manuscript was much improved, and the majority of the comments were well addressed.However, I still feel the rationale for including the CR data is not strong, and I would recommend it be removed. As there was no transcriptomic data for the main study, the CR transcriptome feels especially out of place.

As per the reviewer’s recommendation, we have removed the CR data generated from one of our previous study (Mitchell et al., 2016) to better focus on the narrative of the cross-sectional study.

It would have also been helpful for the authors to show how the metabolomes changes in their cross sectional and CR studies. Do the metabolomes of the CR animals look more like those of the adult animals in the main study?

No further analysis was carried out with this dataset.

In addition, while the authors expanded on their metabolomics interpretations some, they did not correlated their metabolite values to the functional values measured.

In this revised version of the manuscript, we devoted renewed efforts in assessing the strength of the links between metabolites and functional measures (e.g., energetic cost of walking, RER and frailty index) by investigating the organs and serum fine-tuning their metabolism and substrate selection to cope with the increased demand of maintaining health and energy balance. A whole new section devoted to this topic is now included in Results and in the first paragraph of the Discussion. A new set of figures (Figure 6 and associated Figure 6-supplemental figure 1) has been generated as well to support these new findings.